# A hepatic network of dendritic cells mediates CD4 T cell help outside lymphoid organs

Kieran English [1,7], Rain Kwan[1], Lauren E. Holz [2], Claire McGuffog[1], Jelte M. M. Krol [1,8], Daryan Kempe[3], Tsuneyasu Kaisho [4], William R. Heath [2], Leszek Lisowski [5,6], Maté Biro [3], Geoffrey W. McCaughan[1], David G. Bowen [1,9] ✉ & Patrick Bertolino [1,9] ✉

While CD4[+] T cells are a prerequisite for CD8[+] T cell-mediated protection against intracellular hepatotropic pathogens, the mechanisms facilitating the transfer of CD4-help to intrahepatic CD8[+] T cells are unknown. Here, we developed an experimental system to investigate cognate CD4[+] and CD8[+] T cell responses to a model-antigen expressed de novo in hepatocytes and reveal that after initial priming, effector CD4[+] and CD8[+] T cells migrate into portal tracts and peri-central vein regions of the liver where they cluster with type-1 conventional dendritic cells. These dendritic cells are locally licensed by CD4[+] T cells and expand the number of CD8[+] T cells in situ, resulting in larger effector and memory CD8[+] T cell pools. These findings reveal that CD4[+] T cells promote intrahepatic immunity by amplifying the CD8[+] T cell response via peripheral licensing of hepatic type-1 conventional dendritic cells and identify intrahepatic perivascular compartments specialized in facilitating effector T cell-dendritic cell interactions.

CD4[+] helper T cells provide vital signals to conventional CD8[+] cytotoxic T cells through mechanisms collectively known as CD4 help, an important function of CD4[+] T cells that has emerged as a critical factor involved in promoting strong anti-viral and anti-tumor CD8[+] T cell responses[1]. CD4 help to CD8[+] T cells is a highly dynamic process, requiring multiple coordinated cell-cell interactions that ultimately facilitate the transfer of molecular signals to enhance CD8[+] T cell activation, proliferation, and differentiation. The cellular and molecular mechanisms that facilitate CD4 help to CD8[+] T cells can be distinguished into 2 broad categories: (i) CD4 T cell derived cytokines, such as IL-2 and IFN-γ, that act directly or indirectly on CD8[+] T cells;[2–4] and (ii) cognate dendritic cell (DC) licensing, a process in which CD4[+]

T cells recognising antigenic peptides presented by DCs in the context of MHC class II (MHCII) molecules activate DCs via CD40L/CD40 interactions to induce optimal CD8[+] T cell responses[5–7].

Studies in humans and chimpanzees suggest that robust anti-viral CD4[+] T cell responses are a prerequisite for a strong anti-viral CD8[+] cytotoxic T cell response and spontaneous control of hepatitis B virus (HBV) and hepatitis C virus (HCV), which commonly exploit tolerogenic properties of the liver to evade the host immune response and establish chronic infection[8–12]. CD4[+] T cells have also been reported to play an important role in promoting anti-sporozoite CD8[+] T cells responses in the liver during malaria infection[13], anti-tumour CD8[+] T cell immunity in hepatocellular carcinoma (HCC)[14], and pathogenic

[1]Centenary Institute and The University of Sydney, AW Morrow Gastroenterology and Liver Centre, Royal Prince Alfred Hospital, Sydney, NSW, Australia. [2]Department of Microbiology and Immunology at The Peter Doherty Institute for Infection and Immunity, University of Melbourne, Melbourne, VIC, Australia. [3]EMBL Australia, Single Molecule Science node, School of Biomedical Sciences, University of New South Wales, Sydney, NSW, Australia. [4]Department of Immunology, Institute of Advanced Medicine, Wakayama Medical University, Wakayama, Japan. [5]Children's Medical Research Institute, Translational Vectorology Research Unit, Faculty of Medicine and Health, The University of Sydney, Westmead, NSW, Australia. [6]Laboratory of Molecular Oncology and Innovative Therapies, Military Institute of Medicine, Warsaw, Poland. [7]Present address: VIB-UGent Center for Inflammation Research, Ghent, Belgium. [8]Present address: Department of Parasitology, Leiden University Medical Centre, Leiden, The Netherlands. [9]These authors contributed equally: David G. Bowen, Patrick Bertolino. ✉e-mail: d.bowen@centenary.org.au; p.bertolino@centenary.org.au

CD8[+] T cell responses in autoimmune hepatitis and liver transplant rejection[15,16]. However, despite evidence indicating a crucial role for CD4[+] T cells in determining the outcome of liver diseases, studies utilizing murine models to investigate CD4 help to intrahepatic CD8[+] T cells have largely failed to define the mechanisms that facilitate the transfer of CD4 help to CD8[+] T cells specific for antigens expressed in the liver and thus, the cellular and molecular interactions remain poorly understood[17,18].

Here, we induced hepatocyte-specific de novo expression of a model antigen containing CD8[+] and CD4[+] T cell epitopes, known to elicit strong conventional T cell responses, via the use of recombinant adeno-associated viral (rAAV) vector-mediated transduction. By following the response of TCR transgenic (Tg) CD4[+] and CD8[+] T cells recognising these epitopes, we aimed to decipher the cellular and molecular mechanisms that facilitated the transfer of CD4 help. Our findings revealed that, once primed CD8[+] T cells had left secondary lymphoid organs (SLOs), CD4[+] T cells enhanced intrahepatic CD8[+] T cell expansion by promoting local CD8[+] T cell proliferation within portal tracts and peri-central venous (PCV) regions of the liver. CD4 help signals were transferred to intrahepatic CD8[+] T cells via cognate licensing of hepatic cDC1s within the liver itself, in situ in portal tracts and PCV regions. Licensed cDC1s upregulated the expression of cost-imulatory molecules, promoting local reactivation and expansion of intrahepatic CD8[+] T cells, giving rise to a large pool of effector and memory CD8[+] T cells recognising hepatocyte-expressed antigens. These findings indicate that CD4 help via DC licensing can be delivered to CD8[+] T cells at peripheral sites of antigen expression, following priming and egress from SLOs, and identify portal tracts and PCV regions as distinct immune niches within the liver, specialized in facilitating effector T cell-cDC1 interactions and the transfer of CD4 help to intrahepatic CD8[+] T cells. This extra-lymphatic cDC1 licensing serves to expand the CD8[+] T cell response, and may be critical to generate the large number of effector and memory CD8[+] T cells required for clearance and protection against hepatotropic pathogens.

## Results

### CD4 help promotes hepatocyte Ag-specific CD8[+] T cell memory

To address the role of CD4 help in influencing conventional CD8[+] cytotoxic T cell responses to hepatocyte-expressed antigens, we used an rAAV-mediated transduction approach to induce de novo expression of the model antigen "gB-Ag85b" specifically in hepatocytes. gB-Ag85b consists of a membrane bound fusion protein made of two distinct protein fragments containing the immunodominant CD8[+] T cell epitope of the herpes simplex glycoprotein gB (gB$_{498-505}$ or SSIE-FARL) recognised by CD8[+] gBT-I TCR transgenic (Tg) T cells when bound to H2-K[b] [19] and the immunodominant CD4[+] T cell epitope of *Mycobacterium tuberculosis* Ag85b (P25$_{240-254}$ or FQDAY-NAAGGHNAVF) recognised by CD4[+] P25 Tg T cells in the context of I-A[b] [20] (Fig. 1a). rAAV$^{gB-Ag85b}$ is bicistronic and in addition to gB-Ag85b, encodes for green fluorescent protein (GFP), allowing the identification of transduced hepatocytes by flow cytometry and immuno-fluorescence imaging (Fig. 1b, Supplementary Fig. 1a, b). To restrict expression to hepatocytes, gB-Ag85b and GFP expression is driven by a hepatocyte-specific ApoE/hAAT promoter/enhancer element and is packaged in virions containing serotype 8 capsid (Fig. 1a) that displays high hepatotropism in C57BL/6 mice. To confirm that gB-Ag85b expression in hepatocytes resulted in robust cognate CD4[+] and CD8[+] T cell activation, C57BL/6 mice received an adoptive transfer of $1 \times 10^6$ naïve CFSE-labelled gBT-I or P25 T cells and were intravenously inoculated with rAAV$^{gB-Ag85b}$ one day later. rAAV treatment led to the proliferation of the vast majority of gBT-I CD8[+] T cells (Supplementary Fig. 1c), and P25 CD4[+] T cells (Supplementary Fig. 1d) detected in the liver draining lymph nodes at 6-days and 10-days post-rAAV treatment, respectively, indicating robust CD8[+] and CD4[+] T cell activation. Con-sistent with previous results[21,22], titration experiments revealed that a

rAAV$^{gB-Ag85b}$ dose leading to ~1% hepatocyte transduction (Supplemen-tary Fig. 1a, b) resulted in consistently large numbers of memory gBT-I CD8[+] and P25 CD4[+] T cell populations in the liver, spleen, pooled lymph nodes (liver draining and non-draining) and blood at 46-days post-rAAV treatment (Supplementary Fig. 1e, f). This rAAV dose was there-fore used in all subsequent experiments.

To assess whether CD4 help influenced the memory differentia-tion of CD8[+] T cells responding to hepatocyte-expressed antigens, C57BL/6 mice received an adoptive transfer of 10,000 naïve gBT-I CD8[+] T cells and were subsequently treated with either anti-CD4 (GK1.5) antibody to deplete all CD4 T cells, or an isotype control anti-body (Supplementary Fig. 1g–i). Unless otherwise stated, an adoptive transfer of 10,000 naïve Tg T cells was used for all subsequent experiments. Recipient mice were then inoculated with rAAV$^{gB-Ag85b}$, and the functional capability and total number of memory gBT-I CD8[+] T cells activated in the presence or absence of CD4[+] T cells were quantified and compared 46 days post-rAAV inoculation (Fig. 1c). Lymphocytes isolated from recipient livers and spleens were resti-mulated ex vivo with SSIEFARL peptide or media alone and their ability to degranulate and produce cytokines was determined by CD107a degranulation assay and intracellular IFN-γ expression by flow cyto-metry (Figs. 1d–f). The total number of CD107a[+] and CD107a[+]IFN-γ[+] gBT-I CD8[+] T cells was significantly higher in the livers and spleens of isotype control treated mice compared to mice treated with anti-CD4 mAb (Fig. 1e). In the liver, the numbers of CD107a[+] and CD107a[+]IFN-γ[+] gBT-I CD8[+] T cells were approximate 40-fold and 50-fold higher respectively in control versus CD4-depleted mice (Fig. 1e). These dif-ferences were approximately 60-fold and 110-fold in the spleen (Fig. 1e). The same trend was observed when assessing the total number of memory gBT-1 T cells in the pooled LNs and blood (Sup-plementary Fig. 1j). CD4[+] T cells did not significantly increase the proportion of gBT-I CD8[+] T cells that were functional in either liver or spleen (Fig. 1f). Together, these findings indicated that CD4 help plays an important role in expanding the total number of functional memory CD8[+] T cells following the response to hepatocyte-expressed antigen.

### CD4 help enhances intrahepatic CD8[+] T cell expansion

To characterise the mechanisms by which CD4[+] T cells expanded the pool of memory CD8[+] T cells recognising hepatocyte-expressed anti-gen, we next sought to determine when CD4[+] T cells started to influ-ence the CD8[+] T cell response. We first investigated whether CD4[+] T cells enhanced early CD8[+] T cell activation and proliferation in lym-phoid and peripheral compartments, including the liver draining lymph nodes (dLN) and liver, which represent known locations of primary CD8[+] T cell activation following antigen expression by hepatocytes[21]. gBT-1 T cell activation was assessed by quantifying expression of the early activation markers CD69 and CD25, and was detected in both the dLNs and liver as early as 2 days post-rAAV treatment, while most gBT-1 T cells detected in non-liver draining LNs, the spleen, or the blood remained naïve at the same time point (Figs. 2a, b). Across the various tissues, the proportion of CD69[+] gBT-1 cells did not differ significantly between recipient mice treated with isotype control versus anti-CD4 mAb (Figs. 2a, b). CD4[+] T cells did not influence the CFSE dilution profile of gBT-1 T cells in the liver or liver dLNs at 4- or 6-days post-rAAV inoculation (Fig. 2c). Thus, CD4[+] T cells did not boost memory CD8[+] T cell numbers by increasing the pro-portion of cells undergoing primary activation early post-rAAV inoculation, or by accelerating early proliferation during initial prim-ing in lymphoid tissues or the liver.

To determine when the expansion of CD8[+] T cells became dependent on CD4[+] T cells, we quantified the total number of gBT-1 T cells responding to hepatocyte-expressed antigen in rAAV-treated recipient mice containing or lacking CD4[+] T cells at multiple time points from day 5 to 15 post-rAAV treatment. Consistent with the above findings, the total number of gBT-1 T cells at 5-days post-rAAV

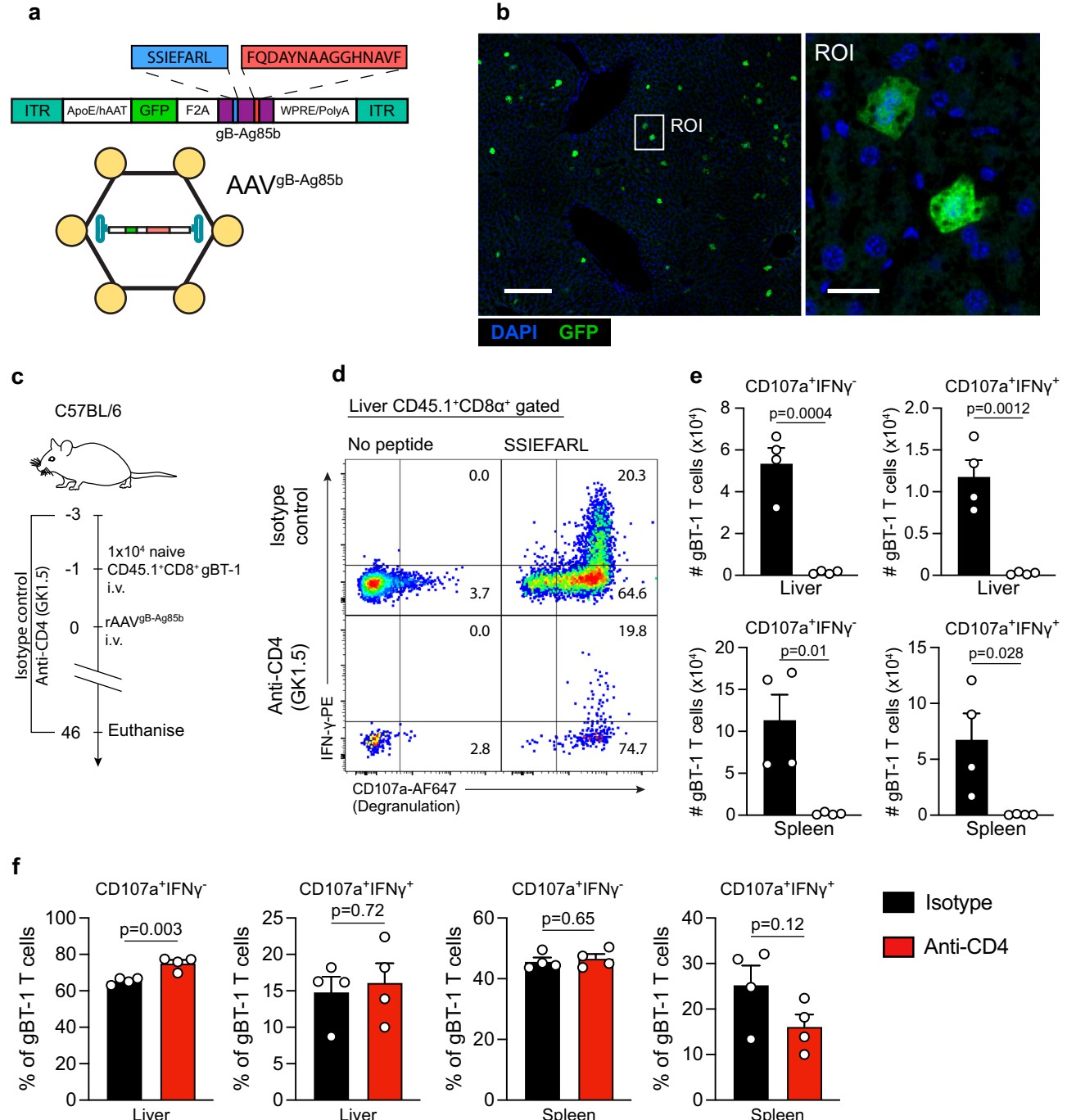

**Fig. 1 | CD4⁺ T cell help expands the number of intrahepatic memory CD8⁺ T cells recognizing hepatocyte-expressed Ag. a** Schematic diagram of the rAAV^gB-Ag85b vector used in this study. **b** Representative confocal image of a C57BL/6 recipient mouse liver at 7-days following rAAV^gB-Ag85b treatment, showing the distribution of GFP⁺ expressing hepatocytes in the liver. A region of interest (ROI) shows a magnified view of transduced hepatocytes. **c** Protocol used to assess the role of CD4 help in intrahepatic CD8⁺ T cell memory generation. **d** Representative FACS plot demonstrating ability of intrahepatic memory CD8⁺ gBT-1 T cells to degranulate and produce IFN-γ after ex vivo peptide stimulation.

**e**, **f** Quantification at 46 days post-rAAV treatment of the total number (**e**) and proportion (**f**) of functional memory CD8⁺ gBT-1 T cells isolated from the livers and spleens of rAAV-inoculated anti-CD4 treated (red) or isotype control treated (black) recipient mice. Data are representative of two independent experiments. $n = 3$ (**b**) or $n = 4$ (**d**–**f**) mice per group. Error bars indicate ± SEM. Analysed with a two-tailed unpaired Student's $t$ test. ns not statistically significant. Scale bars 300 μm/50μm (**b**). Source data are provided as a Source Data file.

treatment were similar in the 2 groups for all tissues investigated (Fig. 2d). CD4 help increased the numbers of gBT-1 T cells in the liver dLNs but not the liver, spleen or blood between 5- and 9-days post-rAAV, however a significant difference was seen in all compartments after 9-days post-rAAV, amounting to 14-, 40- and 25-fold increases in the liver, spleen and blood at 15-days post-rAAV, respectively (Fig. 2d). Significant differences were also observed between CD4⁺ T cell-

depleted and isotype control treated animals when assessing the number of functional intrahepatic and splenic effector gBT-1 T cells at 15-days post-rAAV following ex vivo peptide restimulation (Supplementary Fig. 2a–c), as well as when assessing the number of antigen expressing hepatocytes remaining in the liver at 16-days post-rAAV (Supplementary Fig. 2d, e). Significantly larger numbers of gBT-1 T cells were observed in the liver and spleen compared to the dLNs when

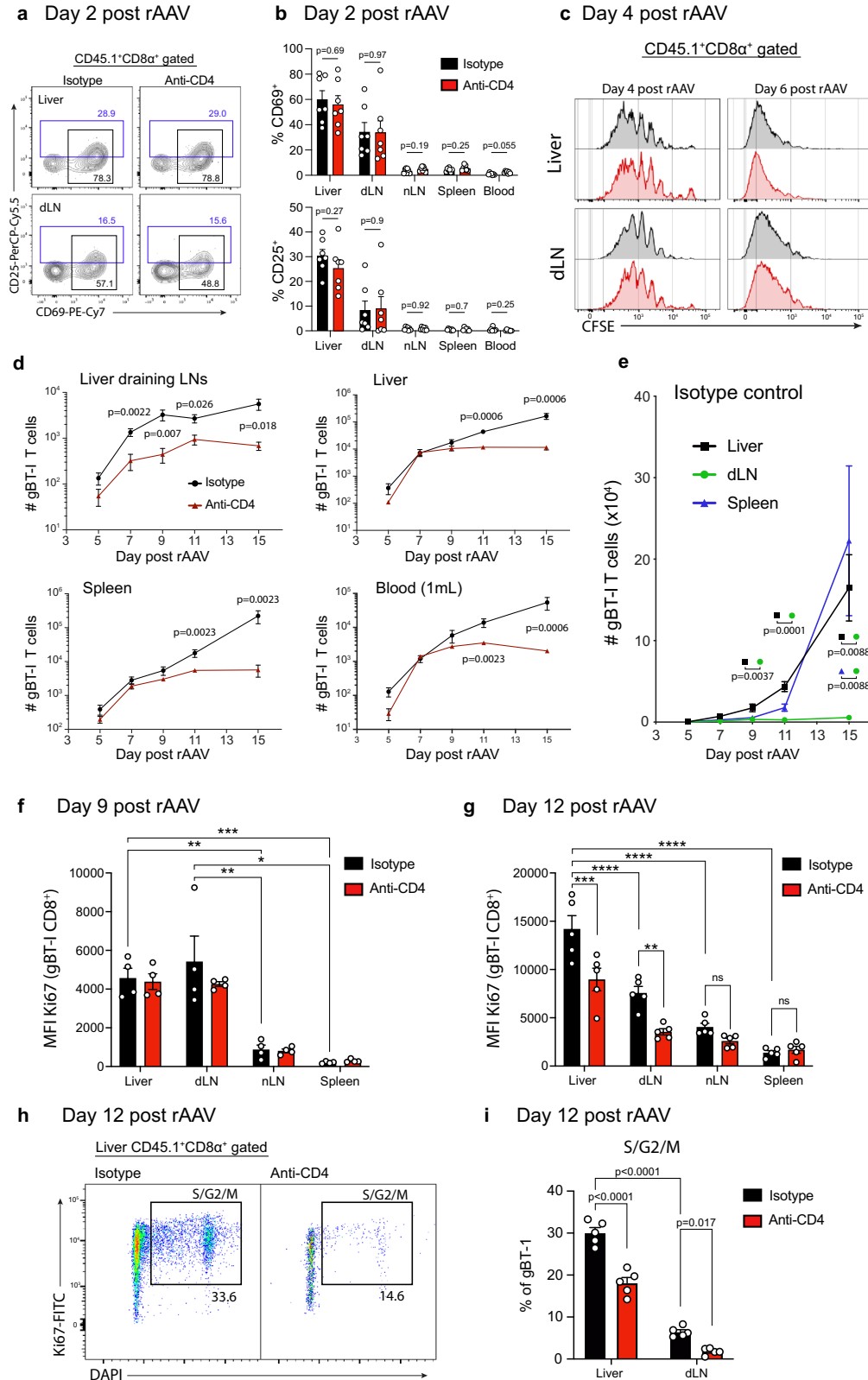

assessing the help-dependent response at day 15 (Fig. 2e), indicating that they were the two main sites where activated CD8[+] T cells accumulated. To determine the compartments in which CD8[+] T cells actively divided, we assessed the proliferative activity of gBT-1 T cells by determining the median fluorescence intensity (MFI) of Ki67 and the proportion of gBT-1 T cells in S, G2 or M stages of the cell cycle[23]. The liver dLNs and the liver contained gBT-1 cells displaying the

highest Ki67 MFI in comparison to the spleen and non-liver dLNs at 9- and 12-days post-rAAV (Figs. 2g, h, Supplementary Fig. 2f, g), suggesting that they were the two main compartments in which CD8[+] T cells actively proliferated. Interestingly, while Ki67 MFI of gBT-1 T cells was similar between the liver and dLNs at day 9, it was significantly increased in the liver at day 12 post-rAAV and the liver contained a higher proportion of cells in S, G2 or M stages of the cell cycle

**Fig. 2 | CD4 help increases the number of effector and memory CD8⁺ T cells by promoting intrahepatic expansion.** **a–c** Activation marker (CD25/CD69) expression by, and proliferation of, transferred gBT-1 CD8⁺ T cells at early time points post-rAAV^gB-Ag85b treatment. Recipient mice received $1 \times 10^6$ CFSE-labelled naïve gBT-1 cells. Representative FACS plots of liver and liver draining LNs (**a**), proportions of recently activated CD69⁺ (top) and CD25⁺ (bottom) gBT-1 cells at 2-days post-rAAV treatment (**b**), and CFSE proliferation profiles of gBT-1 T cells isolated from livers and liver draining LNs of rAAV treated recipient mice (**c**) which had received anti-CD4 mAb or isotype control. **d, e** Kinetics of total gBT-1 T cell numbers in the liver draining LNs, liver, spleen and 1 ml of blood following rAAV^gB-Ag85b treatment of recipient mice. Comparison of total gBT-1 T cells numbers between CD4-depleted and isotype control treated animals (**d**), comparison between the different tissues of isotype control treated animals (**e**). (**f–g**)

Quantification of the level of active cell division (assessed by quantifying the MFI of Ki67) in transferred gBT-1 T cells isolated from CD4 depleted and isotype control treated recipient mice 9-days (**f**) and 12-days (**g**) post-rAAV^gB-Ag85b treatment. **h, i** Representative FACS plot (**h**) and quantification (**i**) of the proportion of gBT-1 T cells isolated from the liver and liver draining LNs (dLN) of CD4 depleted and isotype control treated recipient mice 12-days post-rAAV^B-Ag85b treatment. **b–i** red plots – anti CD4-treated; black plots – isotype control treated. Data are representative of two independent experiments. $n = 7$ mice (**a–e**), or $n = 5$ mice (**f–i**) per group. Error bars indicate ± SEM. Analysed with a two-tailed unpaired Student's $t$ test (**b**) or Mann–Whitney $U$-test (**d, e**) and one-way ANOVA with Tukey's multiple comparisons test (**f–i**), *$p < 0.05$, **$p < 0.01$, ***$p < 0.001$, ****$p < 0.0001$; Source data are provided as a Source Data file.

(Figs. 2g–i), suggesting that gBT-1 T cells exhibited enhanced proliferation in the liver at this later time point. Furthermore, in contrast to day 9, at day 12 post-rAAV the Ki67 MFI of gBT-1 T cells and the proportion in S, G2 or M stages of the cell cycle were higher in mice containing CD4⁺ T cells than in the group lacking CD4⁺ T cells (Figs. 2h, i), suggesting the increased proliferation of CD8⁺ T cells in the liver at this time point was CD4 T cell-dependent. The proportion of gBT-1 T cells in S, G2 or M phases observed in the GI tract, which contains tertiary lymphoid structures including Peyer's patches, was negligible when compared to the liver (Supplementary Fig. 2h), indicating that the help mediated expansion of CD8⁺ T cells observed was relatively confined to the liver and liver draining LNs following hepatocyte antigen expression.

Together, these findings support a model in which CD8⁺ T cell activation and proliferation was initiated in the liver draining lymph nodes and liver following Ag expression by hepatocytes. During the first 9 days post Ag expression, CD4 help induced a modest but consistent increase in the expansion of CD8⁺ T cells in the liver dLNs. However, after 9 days, CD4⁺ T cells enhanced the proliferation of CD8⁺ T cells following migration into the liver, giving rise to a large pool of effector and memory cells.

### Local proliferation in portal tracts and PCV regions drives the help-dependent expansion of CD8⁺ T cells

To investigate the mechanisms facilitating the CD4 help dependent proliferation of CD8⁺ T cells in the liver, we first assessed the anatomical sites where this process occurs. To visualize these events, we imaged the livers of recipient C57BL/6 mice treated with anti-CD4 or isotype control mAb that received adoptive transfer of CD45.1-expressing gBT-1 cells and were treated one day later with rAAV^gB-Ag85b. At 12-days post-rAAV treatment, when the intrahepatic CD8⁺ T cell response expanded in a CD4⁺ T cell-dependent manner (Figs. 2d–h), large clusters of proliferating Ki67⁺ gBT-1 T cells were observed in portal tracts as well as in PCV regions of isotype control treated mice (Fig. 3a top panels, Supplementary Fig. 3a), but not in recipient mice lacking CD4⁺ T cells (Fig. 3a bottom panels; Supplementary Fig. 3b). In the absence of CD4⁺ T cells, the total number of Ki67⁺ gBT-1 T cells within portal tract and PCV clusters was markedly decreased in comparison to isotype control treated mice (Figs. 3b, c), indicating that cluster formation and expansion was dependent on CD4 help. To determine whether large clusters containing proliferating gBT-1 T cells in portal tracts and PCV areas arose from a local clonal expansion of T cells that had migrated into these regions, or alternatively resulted from the accumulation of recirculating dividing T cells, C57BL/6 recipient mice received a mixed population of gBT-1 T cells expressing either tdTomato (9000 cells) or CD45.1 (1,000 cells), one day prior to treatment with a low dose of rAAV^gB-Ag85b (Fig. 3d). At 12-days post-rAAV, gBT-1 T cells expressing either tdTomato or CD45.1 appeared randomly distributed in the sinusoidal regions (Fig. 3e, Supplementary Fig. 3c), while they tended to form homotypic clusters in portal tracts and around central veins (Figs. 3f, g, Supplementary Fig. 3c–e). To

provide more convincing evidence in support of this model, it was important to quantify evidence of gBT-1 clonal expansion in multiple portal tracts/PCV regions, in multiple mice, using an unbiased approach. Thus, we quantified the fraction of nearest non-self (tdTomato⁺ gBT-1 cell) neighbours of CD45.1⁺ gBT-1 cells in sinusoidal regions, portal tracts and PCV regions of the liver. In the sinusoidal compartment, an average of 8.6/10 nearest neighbours of CD45.1⁺ gBT-1 cells were non-self (tdTomato⁺) (Figs. 3e, h, Supplementary Fig. 3c), which resembled the initial adoptive transfer ratio of 9/10 tdTomato⁺:CD45.1 gBT-1 T cells, as well as the overall ratio of tdTomato⁺:CD45.1⁺ gBT-1 T cells detected in the inguinal LNs following expansion and recirculation (Supplementary Fig. 3e), and indicated negligible clonal expansion in this liver compartment. In contrast, in both portal tracts and PCV regions, the fraction of nearest non-self (tdTomato⁺) neighbours of CD45.1⁺ gBT-1 cells was significantly decreased (6.2/10 and 5.8/10, respectively) compared to the sinusoidal compartment (Fig. 3f–h, Supplementary Fig. 3c, d). Quantification of the total number of tdTomato⁺ and CD45.1⁺ gBT-1 T cell in sinusoids, portal tracts and PCV regions showed a ratio resembling that of the initial adoptive transfer, indicating no expansion bias of tdTomato⁺ or CD45.1⁺ gBT-1 T cells in these compartments (Supplementary Fig. 3f). Together, these results indicate that local CD8⁺ T cell expansion contributed significantly to the formation of large Ki67⁺ gBT-1 T cell clusters in portal tracts and PCV regions. This expansion was enhanced by CD4 help, leading to the generation of a larger pool of effector and memory CD8⁺ T cells specific for Ag expressed by hepatocytes.

### Portal tracts and PCV regions support the transfer of CD4 help to intrahepatic CD8⁺ T cells

CD4⁺ T cell-dependent CD8⁺ T cell expansion observed in portal tracts and PCV regions might be the prolonged effect of a differentiation program imprinted by CD4⁺ T cells during CD8⁺ T cell priming. Alternatively, CD4⁺ T cells might have delivered key signals to CD8⁺ T cells at a later stage when both cell subsets were co-localised in portal tracts and PCV regions of the liver. To assess the contribution of these possible pathways, we first sought to determine the period during which the transfer of CD4 help occurred, by depleting CD4⁺ T cells at different time points during the gBT-1 response to rAAV^gB-Ag85b. Depleting CD4⁺ T cells from 3- or 6-days post-rAAV treatment resulted in significant decreases in effector gBT-1 cell numbers in the liver at 15-days post-rAAV treatment when compared to isotype control treatment (Fig. 4a). Interestingly, restricting the availability of CD4 help to the first 9 days after CD8⁺ T cell activation by initiating CD4⁺ T cell depletion at 9-days post-rAAV treatment also resulted in reduced intrahepatic and splenic effector and memory gBT-1 cell numbers at 15-days and 42-days post-rAAV treatment, respectively, compared to isotype control treated mice (Figs. 4b, c, Supplementary Fig. 4a, b). Notably, the number of gBT-1 T cells in the liver following activation in the absence of CD4⁺ T cells from days 3, 6 or 9 were statistically indistinguishable from the number of gBT-1 Tg CD8 T cells present following activation in the absence of CD4⁺ T cells from day 0

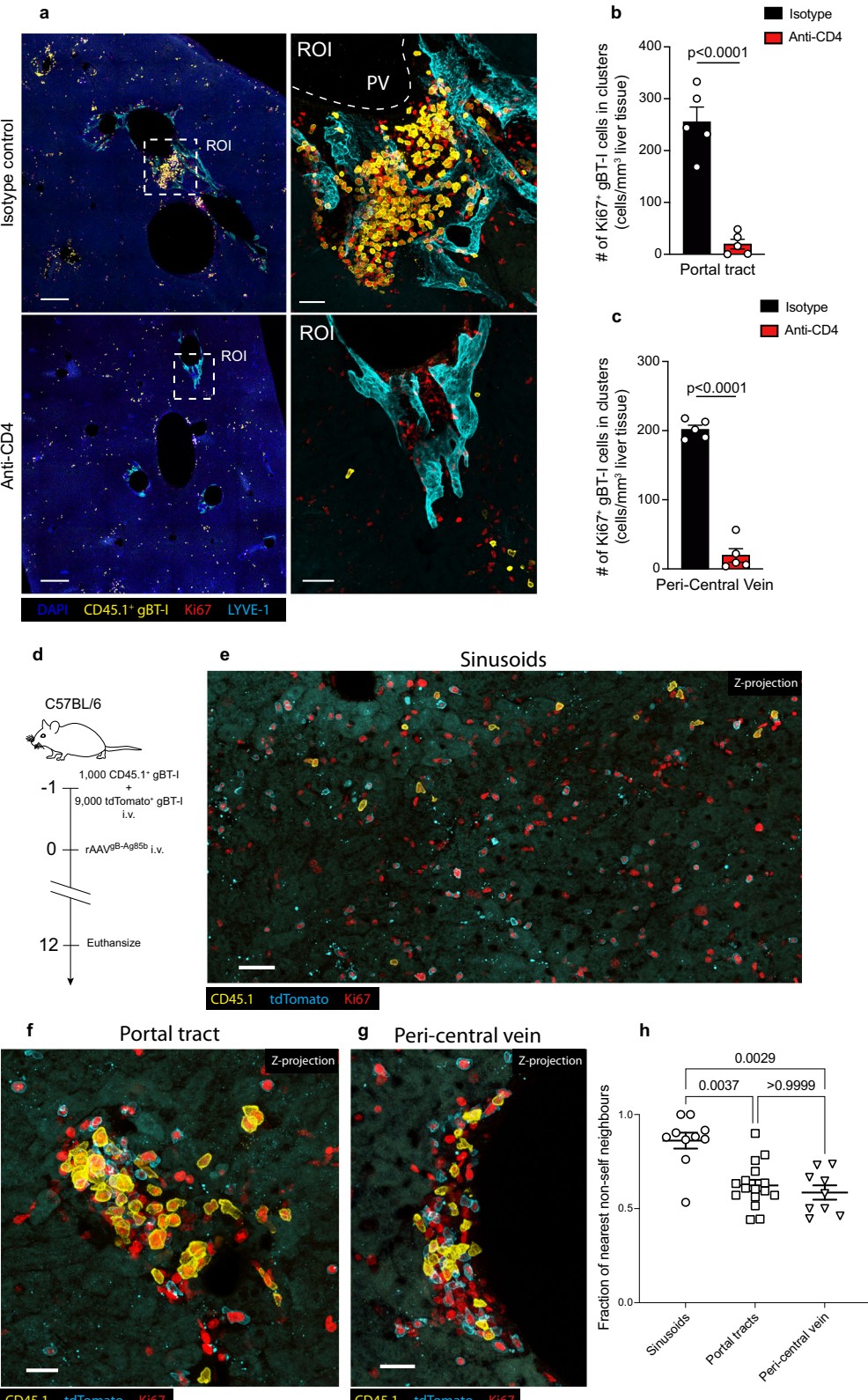

**Fig. 3 | CD4 help-dependent intrahepatic expansion of CD8⁺ T cells is driven by local proliferation in portal tracts and PCV regions. a–c** Intrahepatic localisation of transferred gBT-1 T cells undergoing cell division (Ki67⁺) 12-days following rAAVgB-Ag85b treatment. Representative immunofluorescent (IF) images of livers from isotype control treated (top panels) and CD4 depleted (bottom panels) recipient mice showing the size and location of Ki67⁺ gBT-1 cell clusters (**a**) and quantification of the number of Ki67⁺ gBT-1 T cells in clusters localized to portal tracts (**b**) and peri-central venous (PCV) regions (**c**). **d–h** Intrahepatic gBT-1 T cell clusters detected in portal tracts and PCV regions are the result of local clonal expansion. Protocol (**d**) and representative images showing CD45.1⁺ and tdTomato⁺ gBT-1 T cell clones in the sinusoidal compartment (**e**), portal tracts (**f**) and PCV regions (**g**). Quantification (**h**) of the fraction of nearest non-self (tdTomato⁺) neighbours of CD45.1+ gBT-1 T cells in the sinusoidal compartment, portal tracts and PCV regions. Data are representative of two independent experiments. $n = 5$ mice per group (**a–c**) and $n = 4$ mice per group (d-h). Error bars indicate ± SEM. Analysed with a two-tailed unpaired Student's $t$ test (**b**, **c**) and Kruskal–Wallis test with multiple comparisons (**h**). Scale bars, 300 µm/50 µm (**a**), 40 µm (**e**), 20 µm (**f**, **g**). Source data are provided as a Source Data file.

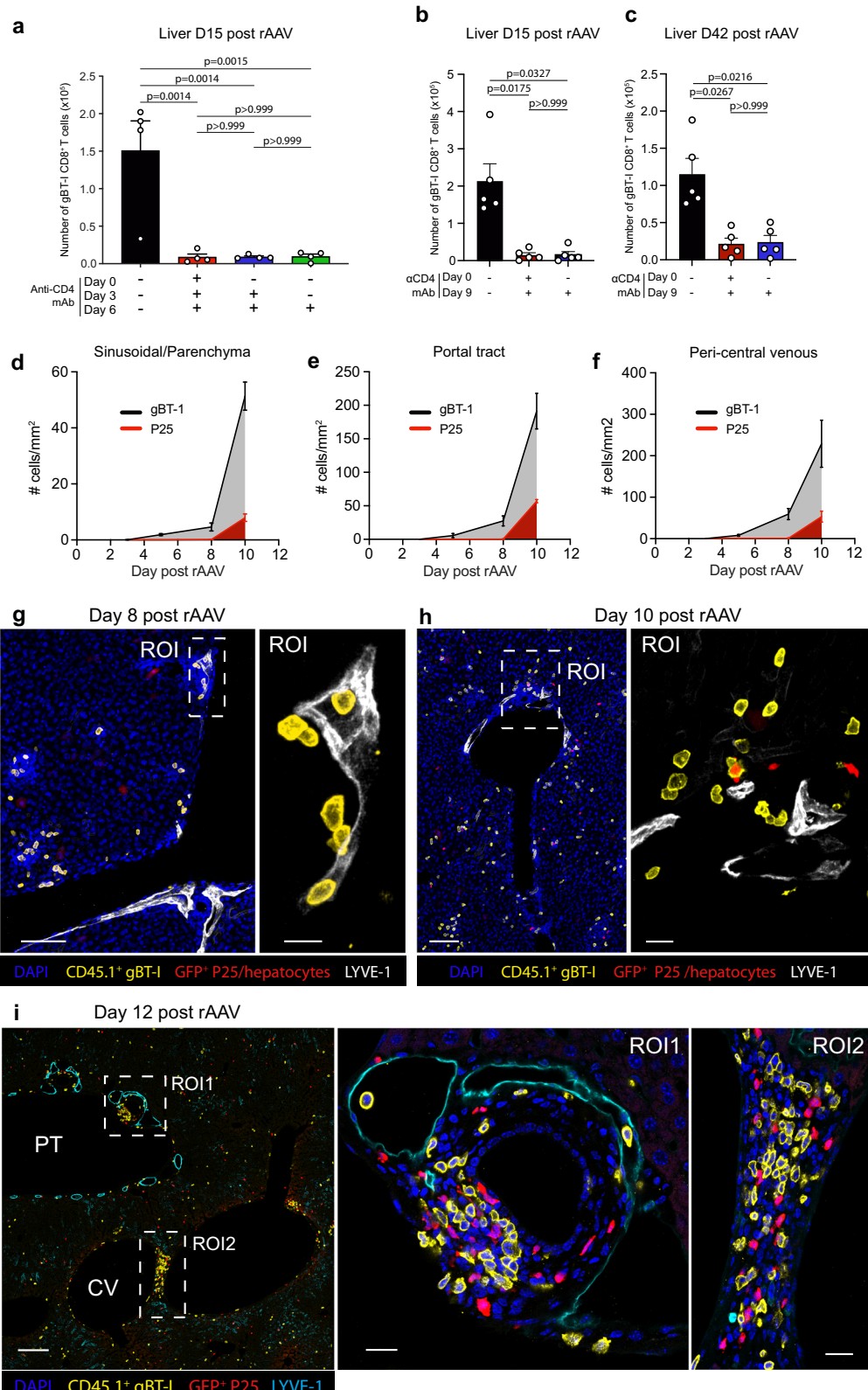

**Fig. 4 | Portal tracts and PCV regions support the transfer of CD4 help to intrahepatic CD8+ T cells. a–c** Total number of intrahepatic effector and memory gBT-1 T cells post-rAAV treatment of recipient mice administered isotype control, or in which CD4 T cells were depleted at day 0, 3 or 6 (**a**), or at day 0 or 9 (**b, c**). **d–f** Kinetics of adoptively transferred gBT-1 and P25 T cell numbers detected in parenchyma/sinusoidal (**d**), portal tract (**e**) and PCV (**f**) liver compartments following rAAV treatment of recipient C57BL/6 mice (red P25 T cells; black/grey gBT-1 T cells). **g–i** Representative IF images of gBT-1 and P25 T cells in the liver of rAAV treated animals 8- (**g**), 10- (**h**) and 12- (**i**) days post-rAAV treatment, showing that CD4+ T cells enter portal tracts and interact with cognate CD8+ T cells after day 8. Data are representative of two independent experiments. *n* = 4 (**a**), *n* = 5 (**b, c**), *n* = 4 (**d–i**) mice per group. Error bars indicate ± SEM. Analysed with Kruskal–Wallis test with multiple comparisons. Scale bars, (**g**) 100 μm/10 μm, (**h**) 100 μm/20 μm, (**i**) 150 μm/20 μm/20 μm. Source data are provided as a Source Data file.

(Figs. 4a–c). These results suggested that CD4⁺ T cell-dependent CD8⁺ T cell expansion was not the result of a differentiation program imprinted during priming, but was rather induced by key signals provided by CD4⁺ T cells after 9 days post-rAAV.

To investigate whether CD4⁺ T cells transferred cognate signals to CD8⁺ T cells within portal tracts and PCV regions, naive gBT-1 and P25 T cells were adoptively co-transferred into recipient mice that were treated with rAAV^gB-Ag85b one day later. Recipient livers were imaged at different time points post-rAAV administration to assess the intrahepatic migration kinetics of gBT-1 and P25 T cells, and to pinpoint the time period during which cognate interactions between these two T cell subsets might occur. At 3-days post-rAAV treatment, small numbers of gBT-1 cells were observed in the liver sinusoids (Supplementary Fig. 4c). Donor gBT-1 cells were first detected within portal tracts and PCV regions at 5-days post-rAAV inoculation and their numbers progressively increased between 3- and 8-days post-rAAV treatment (Figs. 4d–f). CD4⁺ P25 T cells were not detected in the liver until 8-days post-rAAV, when small but consistent numbers were found in the sinusoid but not portal tracts or PCV regions (Fig. 4g, Supplementary Fig. 4d). P25 T cells started to be consistently detected in portal tracts and PCV regions at 10-days post-rAAV treatment, where they could be observed in the proximity of gBT-1 T cells (Fig. 4h, Supplementary Fig. 4e). Consistent with results described above, large clusters of gBT-1 T cells were observed in portal tracts and PCV regions at 12-days post-rAAV treatment (Fig. 4i). At that time, T cell clusters in portal tracts and PCV regions contained abundant gBT-1 and P25 T cells, within which gBT-1 T cells were, on average, 4-fold closer to P25 T cells compared to the sinusoidal compartment (Fig. 4i, Supplementary Fig. 4f). The migration of CD4⁺ T cells into portal tracts and PCV regions coincided with the period during which CD4 help transfer occurred and preceded the help-dependent expansion of CD8⁺ T cells in portal tracts and PCV regions. As we have previously shown that in this model, CD4⁺ T cells responding to antigens expressed de novo by hepatocytes are predominantly activated in the liver draining celiac and portal lymph nodes[22], these findings suggests that CD4⁺ T cells, initially primed in the liver draining LNs, migrated into portal tracts and PCV regions where they helped CD8⁺ T cells and induced local intrahepatic CD8⁺ T cell expansion.

## Intrahepatic CD4 help depends on CD40-CD40L-mediated DC licensing

As CD4 help to intrahepatic CD8⁺ T cells can be delivered in the form of CD4⁺ T cell-derived cytokines or by CD40L-CD40 mediated DC licensing[1], we next assessed the contribution of CD40-CD40L dependent licensing of DCs in facilitating the help-dependent intrahepatic expansion of CD8⁺ T cells. gBT-1 T cells were transferred into C57BL/6 recipient mice that were subjected to either CD4⁺ T cell depletion, CD40L blockade, or isotype control antibody treatment, starting one day prior to rAAV inoculation and lasting until sacrifice. Recipient mice were treated with rAAV^gB-Ag85b, and the accumulation of intrahepatic gBT-1 effector cells was assessed 15-days later. Blocking CD40L-CD40 interactions resulted in a marked decrease in the accumulation of intrahepatic effector gBT-1 cells compared to isotype control treatment (Fig. 5a). The reduced numbers of intrahepatic gBT-1 cells after CD40L blockade commenced prior to antigen expression were similar to numbers following CD4⁺ T cell depletion (Fig. 5a), indicating that CD40L-CD40 interactions were a prerequisite for the help-dependent expansion of CD8⁺ T cells responding to hepatocyte-expressed antigen. Initiating CD40L blockade at day 9 post-rAAV treatment also led to decreased numbers of effector gBT-1 cells in the liver at 15-days post-rAAV treatment when compared to isotype control treatment (Fig. 5b), confirming that CD40L-CD40 interactions were required during the same time window during which CD4⁺ T cells transferred their help. In vivo stimulation of CD40 signalling in CD4-depleted mice at day 9 post-rAAV treatment using agonistic anti-CD40 mAb resulted

in an approximate 140-fold increase in the total number of intrahepatic effector gBT-1 T cells compared to CD4-depleted mice treated with an isotype control, indicating that artificially providing CD40 signalling after day 9 post-rAAV could simulate the transfer of CD4 help to intrahepatic CD8⁺ T cells (Fig. 5c). We, and others, have recently highlighted that portal tracts and PCV regions represent distinct immune niches within the steady state liver, enriched with DCs ideally positioned to interact with intrahepatic lymphocytes migrating between the blood and lymphatic circulations[24,25]. Having established that CD8⁺ T cell expansion was dependent on CD4 T cell help involving CD40L-CD40 interactions, we next sought to determine whether CD40 expressing professional APCs were positioned to facilitate this process in portal tracts and PCV regions. We adoptively transferred gBT-1 and P25 T cells into C57BL/6 recipient mice one day before treatment with rAAV^gB-Ag85b and imaged the livers 12 days later. gBT-1 and P25 T cells clustered with MHCII^high cells that were closely associated with LYVE-1⁺ lymphatic vessels within portal tracts (Fig. 5d), as well as MHCII^high cells localised within PCV regions (Supplementary Fig. 5a). Importantly, both gBT-1 and P25 T cells were, on average, 6- and 9-fold closer to MHCII⁺ cells, respectively, in these regions compared to the sinusoidal compartment (Supplementary Fig. 5b, c). gBT-1, P25 and MHCII^high cells were readily observable in all portal tracts and PCV clusters (Supplementary Fig. 5a), suggesting that professional APCs located in portal tracts closely interacted with CD4⁺ T cells and CD8⁺ T cells and facilitated the transfer of cognate CD40-CD40L-mediated CD4 help within these compartments.

## Limited role of lymph nodes in late intrahepatic CD8⁺ T cell expansion

The above observations suggested that cognate CD40-CD40L-mediated CD4 help could be delivered within the liver during the period of gBT-1 T cell expansion. However, provision of help within lymph nodes at later time-points with subsequent migration of activated gBT-1 T cells to the liver could not be excluded as a dominant contributor. To further explore the role of T cells exiting the lymph nodes to gBT-1 cells expanding within the liver, we abrogated T cell egress from lymph nodes via treatment of recipient C57BL/6 mice containing gBT-1 T cells with FTY720 (fingolimod) beginning from 2 days post-rAAV inoculation, and analyzed the total accumulation of intrahepatic and splenic effector gBT-1 cells and the proliferation of intrahepatic gBT-1 T cells in portal tracts and PCV regions (Fig. 6a). FTY720 treatment beginning at 2-days post-rAAV inoculation induced a severe loss of gBT-1 CD8 T cells circulating in the blood in comparison with vehicle treated control mice (Supplementary Fig. 6a). Compared to vehicle control, FTY720 treatment beginning at 2-days post-rAAV inoculation resulted in 20-fold and 65-fold decreases in the numbers of intrahepatic and splenic effector gBT-1 T cells, respectively (Fig. 6b, Supplementary Fig. 6b). In the liver, the numbers of Ki67⁺ gBT-1 T cells in portal tract and PCV clusters were significantly decreased by 45-fold and 10-fold, respectively (Figs. 6c, d). These findings were consistent with the thesis that initial priming in lymph nodes was required for significant expansion and accumulation of CD8⁺ T cells responding to hepatocyte-expressed antigen within the liver. To assess the contribution of liver draining lymph nodes to the increased numbers of intrahepatic gBT-1 T cells after day 9, we performed similar experiments to those described above except that FTY720 treatment was started at day 9 post-rAAV inoculation instead of day 2 (Fig. 6e). In contrast to FTY720 treatment that started at day 2, starting FTY720 treatment at 9-days did not influence the total number of intrahepatic and splenic effector gBT-1 T cells (Fig. 6f, Supplementary Fig. 6c) nor the numbers of Ki67⁺ gBT-1 T cells in portal tract and PCV clusters (Figs. 6g, h, Supplementary Fig 6e). This latter result suggested that egress of T cells from lymph nodes was not required for the help-dependent intrahepatic expansion of CD8⁺ T cells once activated CD8⁺ and CD4⁺ T cells had

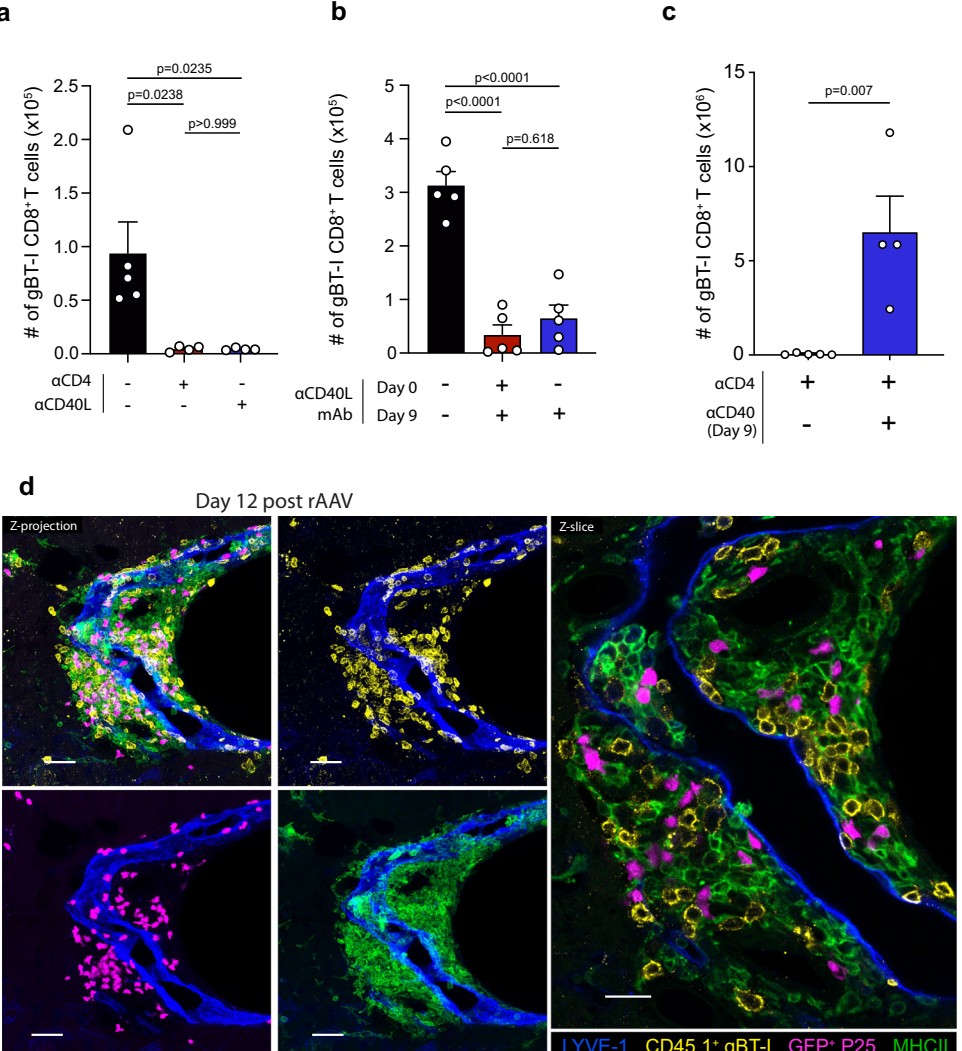

**Fig. 5 | CD4$^+$ T cells promote the intrahepatic expansion of CD8$^+$ T cells via CD40-CD40L mediated APC licensing. a** Comparison of the total number of intrahepatic effector gBT-1 T cells at 14 days post-rAAV treatment in recipient mice treated with anti-CD4 depleting, anti-CD40L blocking Ab, or isotype control Ab. **b** Comparison of the total number of intrahepatic effector gBT-1 T cells at 15 days post-rAAV treatment of recipient mice treated with isotype control, or CD40L blocking Abs beginning day 0 or day 9 post-rAAV. **c** Comparison of the total number of intrahepatic effector gBT-1 T cells at 15 days post-rAAV treatment of CD4-depleted recipient mice administered agonist anti-CD40 mAb or isotype control 9-days post-rAAV. **d** Representative IF image of a portal tract cluster containing transferred gBT-1 and P25 T cells interacting with MHCII$^{high}$ cells in the liver 12-days post-rAAV treatment of recipient mice. Data are representative of two independent experiments. $n = 4$ (**a**) or $n = 5$ (**b**), $n = 4$ or 5 (**c**), $n = 3$ mice per group. Error bars indicate ± SEM. Analysed with one-way ANOVA with Tukey's multiple comparisons test (**a**, **b**), or a two-tailed Student's $t$ test (**c**). Scale bars, 40 μm/20 μm (**d**). Source data are provided as a Source Data file.

trafficked to the liver, consistent with the provision of CD4 help at 9 days and beyond within the liver itself.

### In situ licensing of hepatic cDC1s promotes intrahepatic CD8$^+$ T cell expansion

Finally, we investigated the cellular population facilitating the transfer of cognate CD40-CD40L mediated CD4 help to intrahepatic CD8$^+$ T cells in portal tracts and PCV regions. Given the rich network of cDC1s associated with the portal tract lymphatic vasculature and PCV regions in the steady state liver[24,25], we hypothesized that they represented the primary professional APC licensed by CD4$^+$ T cells to promote the help-dependent expansion of intrahepatic CD8$^+$ T cells. To explore whether CD4$^+$ T cells, CD8$^+$ T cells and hepatic cDC1s established the cell-cell interactions necessary to facilitate the transfer of cognate CD4 help in portal tracts and PCV regions, gBT-1 and P25 T cells were transferred into XCR1$^{Venus/+}$ reporter mice[26] that received rAAV$^{gB-Ag85b}$ inoculation one day later. 12-days post-rAAV inoculation, livers of recipient mice were imaged by confocal microscopy. All clusters of gBT-1 and P25

T cells in portal tract and PCV regions included XCR1$^+$ cells (Figs. 7a–c, Supplementary Fig. 7a, b) and both gBT-1 and P25 T cells were, on average, 4-5-fold closer to XCR1$^+$ cells in these regions compared to the sinusoidal compartment (Figs. 7d, e). Direct cell-cell contacts between the 3 cell types were readily observed in portal tracts and PCV regions (Fig. 7c, Supplementary Fig. 7b). Together, these results indicate that CD4$^+$ T cells, CD8$^+$ T cells and hepatic cDC1s established the interactions required for the transfer of cognate CD4 help in portal tracts and PCV regions.

To determine whether hepatic cDC1s were capable of cross-presenting and/or presenting hepatocyte-derived antigens to CD8$^+$ and CD4$^+$ T cells within the liver, we sorted XCR1$^+$ cDC1s from livers during the expansion phase of the help-dependent CD8$^+$ T cell response, at 12-days post-rAAV$^{gB-Ag85b}$ treatment, and assessed their ability to initiate the activation and proliferation of naïve CTV-labelled gBT-1 or P25 T cells in co-culture. XCR1$^+$ cDC1s isolated from the livers of rAAV treated mice induced activation and proliferation of naïve CD8$^+$ gBT-1 and CD4$^+$ P25 T cells following 72 h of co-culture, as

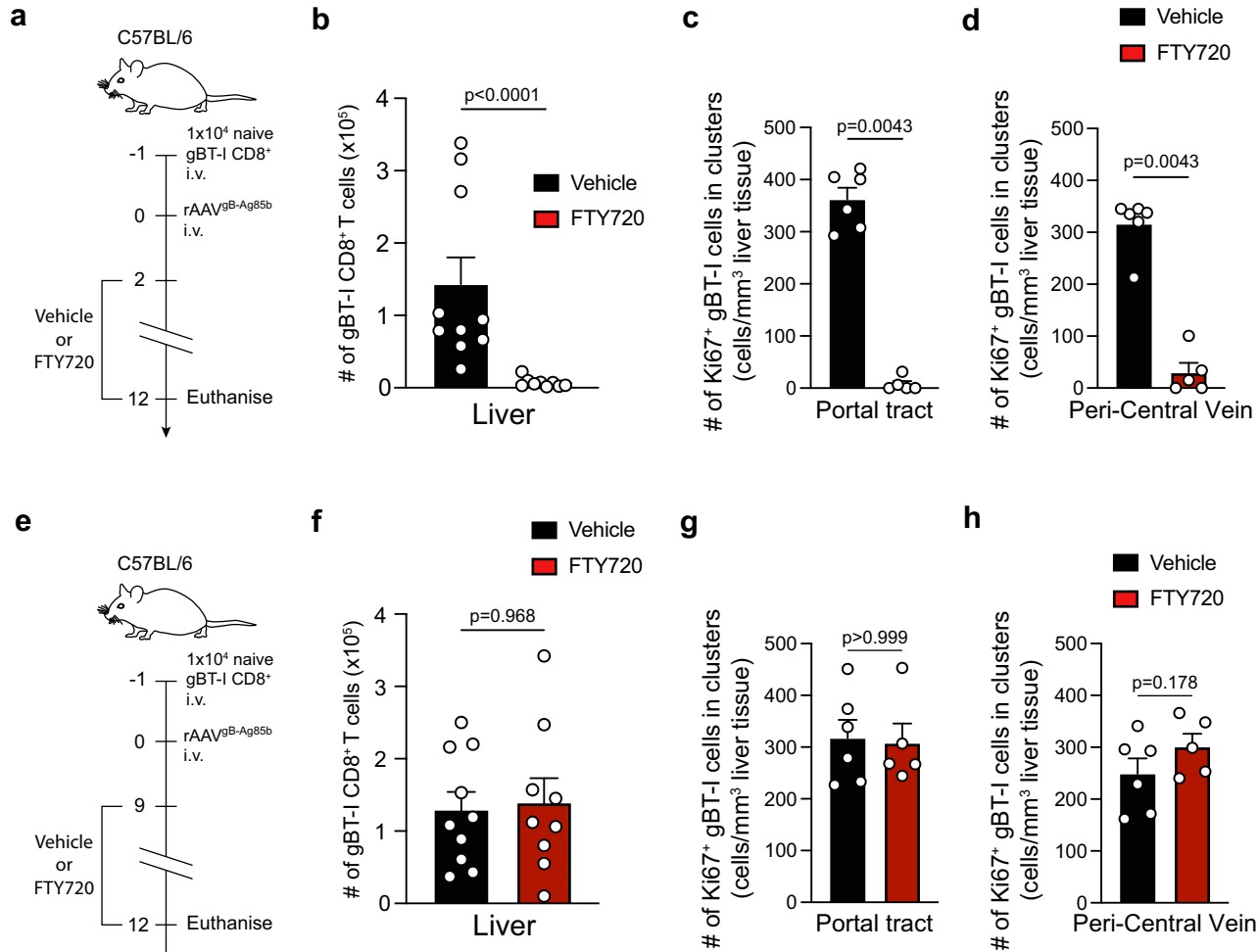

**Fig. 6 | Kinetics of requirement for T cell lymph node egress for help-dependent intrahepatic CD8⁺ T cell expansion. a–d** Comparison between C57BL/6 recipient mice treated with vehicle control (black) or FTY720 (red) beginning 2-days post-rAAV. Protocol (**a**); quantification of the total number of intrahepatic gBT-1 T cells (**b**) and of Ki67⁺ gBT-1 T cells numbers that form clusters in portal tracts (**c**) and PCV(**d**) liver regions at 12-days post-rAAV treatment. **e–h** Comparison between C57BL/6 recipient mice treated with vehicle control (black) or FTY720 (red) beginning 9-days post-rAAV. Protocol (**e**) and quantification of the total number of intrahepatic gBT-1 T cells (f) and Ki67⁺ gBT-1 T cells forming clusters in portal tract (**g**) and PCV regions (**h**) at 12-days post-rAAV treatment. Data are representative of two independent experiments. n = 9 or 10 (**b**), n = 5 or 6 (**c**, **d**), n = 9 or 10 (**f**), 5 or 6 (**g**, **h**) mice per group. Error bars indicate ± SEM. Analysed with a two-tailed unpaired Mann–Whitney U-test. Source data are provided as a Source Data file.

indicated by upregulation of CD44 and dilution of CTV (Figs. 7f, g). Importantly, XCR1⁺ cDC1s isolated from the livers of rAAV-treated mice administered CD40L blocking mAb showed decreased ability to activate naïve gBT-1 T cells following 72-h of ex vivo co-culture compared to those from animals treated with isotype control antibody (Figs. 7h, i). Flow cytometric analysis revealed a decrease in the expression of co-stimulatory molecules CD86 and CD40 on XCR1⁺ cDC1s isolated from the livers of anti-CD4 depleting mAb treated mice 12-days following rAAV^gB-Ag85b treatment, when compared to isotype control treated mice (Fig. 7j, Supplementary Fig. 7c). Similarly, image analysis revealed that CD103⁺MHCII^high hepatic cDC1s in portal tracts exhibited lower expression of CD40 following anti-CD4 mAb treatment compared to isotype control (Supplementary Fig. 7d, e). Together, these results suggested that CD40-CD40L-mediated licensing of hepatic cDC1s by CD4⁺ T cells in situ is required for the help-mediated expansion of CD8⁺ T cells in the portal tracts and PCV regions.

Finally, to determine whether hepatic cDC1s were required for help-dependent proliferation of CD8⁺ T cells in portal tracts and PCV regions, gBT-1 T cells were transferred into XCR1^DTRVenus/+ mice or XCR1^+/+ littermate controls[26] one day prior to administration of rAAV^gB-Ag85b (Supplementary Fig. 7f). Recipient XCR1^DTRVenus/+ and

XCR1^+/+ mice were then treated with diphtheria toxin (DTx) 9-days post-rAAV administration to deplete XCR1⁺ DCs during the intrahepatic expansion phase of the CD4 help-dependent CD8⁺ T cell response, but not during initial T cell priming. Depletion of XCR1⁺ cDC1s at 9-days post-rAAV treatment markedly reduced the size and number of Ki67⁺ gBT-1 cell clusters within portal tracts and PCV regions (Figs. 7k, l, Supplementary Fig. 7g). Furthermore, effector CD8⁺ gBT-1 T cells failed to accumulate in the liver and spleen in mice depleted of XCR1⁺ cDC1s at 9-days post-rAAV inoculation (Figs. 7m, n).

Together, these results suggest that during the response to hepatocyte-expressed antigen, hepatic cDC1s are licensed by CD4⁺ T cells via CD40L-CD40 interactions within portal tracts and PCV regions, which in turn enhances the local proliferation of CD8⁺ T cells and ultimately promotes the generation of a large pool of effector and memory CD8⁺ T cells.

## Discussion

While CD4⁺ T cells are known to have an important role in generating strong CD8⁺ T cell-mediated immunity against liver pathogens and tumours, their precise influence on intrahepatic effector and memory CD8⁺ T cells have remained elusive, as have the mechanisms that

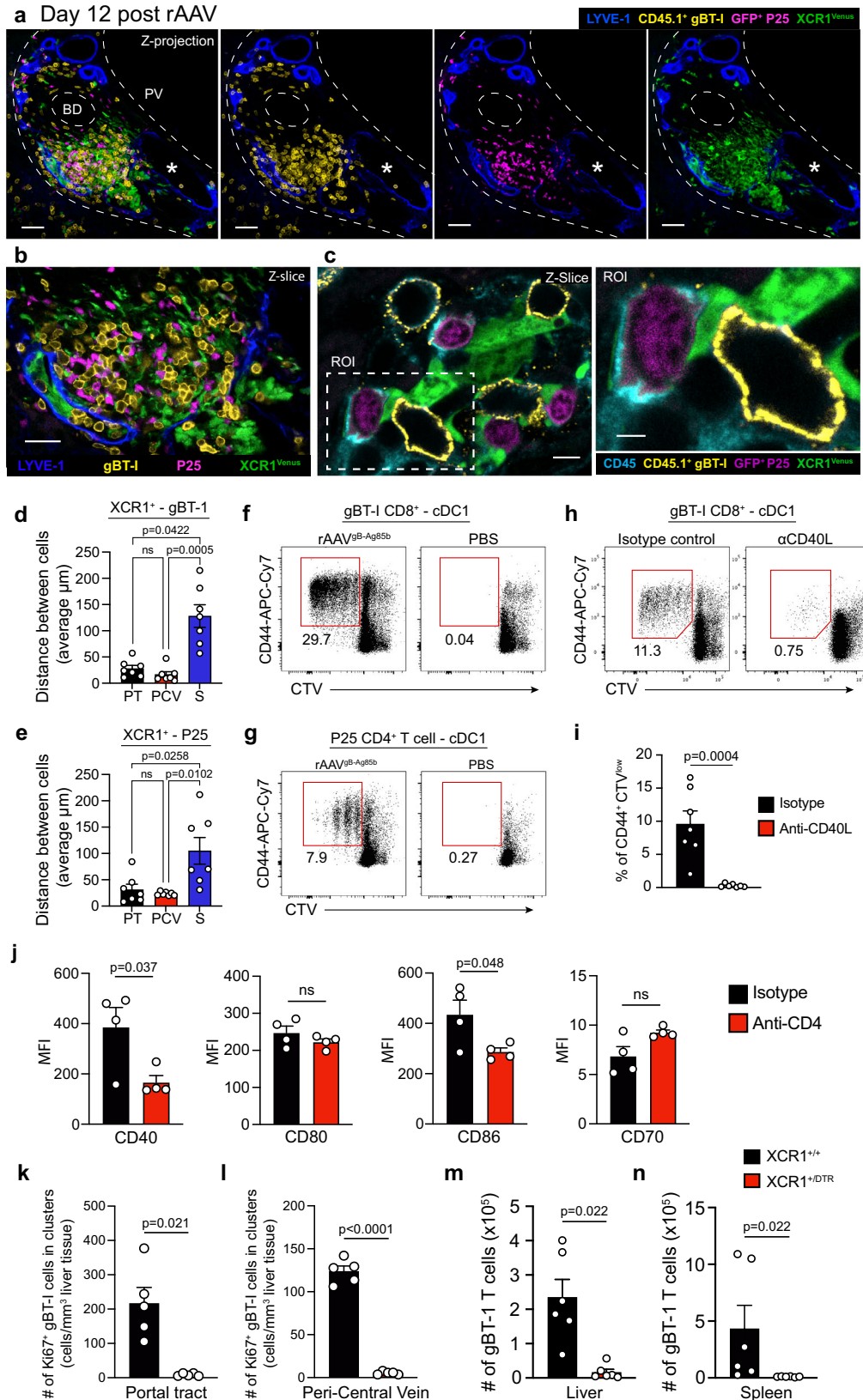

facilitate the transfer of CD4 help to intrahepatic CD8+ T cells. By developing a mouse model in which TCR transgenic CD4+ and CD8+ T cells respond to the same antigen expressed by hepatocytes, this study demonstrates that the main role of CD4+ T cells is to increase the total number of effector and memory CD8+ T cells activated against hepatocyte-derived antigens. Surprisingly, CD4+ T cells achieve this by

licensing hepatic XCR1+ cDC1s in situ within portal tracts and peri-central venous regions via CD40L-CD40 interactions. Licensed cDC1s subsequently encounter effector CD8+ T cells migrating through portal tracts and enhance extra-lymphoid (intrahepatic) clonal expansion. This amplification of the intrahepatic CD8+ effector T cell response serves to generate a large pool of memory cells capable of rapidly

**Fig. 7 | Liver cDC1s cross-presenting hepatocyte-expressed Ag are licensed in situ to promote the intrahepatic expansion of CD8⁺ CTLs. a–c** Representative confocal IF images of the liver of XCR1$^{Venus/+}$ rAAV-treated recipient mice at 12-days post rAAV, showing large clusters of XCR1⁺ cDC1s, gBT-1 and P25 T cells in PT regions Scale bars 50 μm (**a**), 30 μm (**b**), 4 μm/2 μm (**c**). **d, e** Quantification of the average distance (μm) separating XCR1⁺ cells with gBT-1 T cells and (**d**), or with P25 T cells (**e**) in portal tracts (PT), peri-central vein (PCV) and sinusoidal (S) liver regions. Data points represent a single region of interest from $n = 7$ mice from 2 independent experiments. **f, g** Hepatic XCR1⁺ cDC1 isolated from recipient mice treated, 12-days prior, with rAAV$^{gB-Ag85b}$ or PBS, were assessed for their ability to activate and induce proliferation (assessed by CD44 upregulation and CTV dilution), of naïve gBT-1 (**f**) or P25 (**g**) T cells co-cultured for 72 h. **h, i** Same as (**f, g**) but recipient mice were also treated with isotype control (black) or anti-CD40L blocking mAb (red) to assess the role of CD40/CD40L interactions. Representative flow plots (**h**), and proportion of proliferating CD44$^{high}$CTV$^{low}$ gBT-1 T cells (**i**). **j** Co-

stimulatory molecule expression levels in hepatic cDC1s isolated from rAAV$^{gB-Ag85b}$–treated mice that also received isotype control (black) or anti-CD4 depleting mAb (red) at 12-days post rAAV. **k, l** Quantification of Ki67⁺ gB T-1 T cell numbers forming clusters in PT (**k**) and PCV regions (**l**) at 12-days post-rAAV treatment in XCR1$^{+/+}$ (black) and XCR1$^{DTR/+}$ (red) recipient mice treated with DTx, 9-days prior. **m, n** Quantification of total numbers of intrahepatic (**m**) and splenic (**n**) effector gBT-1 cell in recipient mice treated with rAAV, 15-days prior. Comparison between diphtheria toxin treated XCR1$^{+/+}$ (black) and XCR1$^{DTR/+}$ (red) recipient mice at 9-days post-rAAV. Data representative of two independent experiments. $n = 5$ (**a–c**), $n = 7$ (**d, e**), $n = 3$ or 4 (**f, g**), $n = 7$ (**i**), $n = 4$ (**j**) $n = 5$ (**k, l**) and $n = 6$ (**m, n**) mice per group. Error bars indicate ± SEM. Analysed with a Kruskal-Wallis test with multiple comparisons (**d, e**), a two-tailed unpaired Student's *t* test (**i, j**) or a two tailed unpaired Mann−Whitney *U* test (**k−n**); ns not statistically significant. Source data are provided as a Source Data file.

acquiring effector functions following antigen re-encounter. In addition to identifying an important role for cDC1 licensing in promoting robust intrahepatic CD8⁺ T cell immunity, these findings challenge the paradigm that cognate DC licensing is restricted to SLOs[1,27,28], and suggest that these molecular signals, critical for host protection against intracellular pathogens and tumours, can also be transferred to CD8⁺ T cells in a peripheral tissue.

Our findings are consistent with studies identifying XCR1⁺ cDC1s as the essential bridge between the CD4⁺ and CD8⁺ T cells that facilitate the transfer of CD4 help via cognate DC licensing[27–30]. Effective T cell responses are initiated by cDCs in SLOs, the microanatomy of which are adapted to increase the chance that naïve CD8⁺ and CD4⁺ T cells, whose precursor frequency could be as low as 500 out of a total of 50–100 billion[31,32], encounter APCs presenting their cognate peptide antigens on MHC molecules. Thus, it is not surprising that a critical role for cDC1 licensing was first demonstrated in SLOs during the initial events of a T cell response[27,28,33]. In contrast, despite recent studies demonstrating an important role for extra-lymphoid CD8⁺ T cell-cDC cross-talk in maintaining robust anti-viral and anti-tumour immunity[34–36], studies investigating the role of cognate DC licensing in non-lymphoid peripheral tissues, once activated T cells have left SLOs, have been lacking. Our data reveal that in addition to providing help via DC licensing in SLOs during the initial stages of a T cell response, CD4⁺ T cells are able to transfer their help to CD8⁺ T cells via cognate licensing of cDC1s at a peripheral site of antigen expression, following T cell egress from SLOs. Additionally, our findings indicate that portal tracts and PCV regions represent distinct immune niches within the liver, specialized in facilitating effector T cell-cDC1 interactions, including those necessary for the transfer of CD4 help to intrahepatic CD8⁺ T cells.

The portal interstitium surrounds the branches of the portal vein, hepatic artery and biliary tree that extend throughout the entire liver; thus, it occupies a significant volume and represents a compartment extending throughout this large solid organ[37]. Although widely thought to represent an immunologically inert conduit composed of connective tissue and fibroblasts in which lymphocytes were carried with the flow of lymph, the portal interstitium is now known to be a complex and organised compartment rich in MHCII$^{high}$ myeloid cells that include cDC1s, cDC2s and at least two non-Kupffer cell macrophage populations[24,25]. Thus, like SLOs, the portal tracts are a confined compartment containing large numbers of professional APCs, including cDC1s, that are localised to areas of high CD4⁺ and CD8⁺ T cell traffic. These properties suggest that portal tracts represent an important immunological site within the liver, possessing a unique ability to orchestrate intrahepatic T cell immunity. This view of the portal tract compartment provides new insight into the liver's paradoxical immune properties, the ability to induce tolerance while remaining the target of effective immune responses. The mechanisms regulating the balance of tolerance and immunity in this organ remain

the subject of intense debate and the underlying mechanisms remains unresolved[38]. Most studies support the view that intrahepatic tolerance is the resulting outcome of defective primary or secondary T cell activation in the hepatic sinusoids by unconventional APCs residing in these areas, including liver sinusoidal endothelial cells (LSECs), hepatocytes, Kupffer cells (KCs) and hepatic stellate cells[38]. In this context, a corollary resulting from the findings of the present study is that the liver includes distinct compartments with opposing immunological properties. The first compartment is comprised of the liver lobules supplied by the hepatic sinusoids, and is functionally biased towards the induction of CD8⁺ T cell tolerance. Portal tracts represent the second compartment, serving to amplify the intrahepatic CD8⁺ T cell response by supporting the local licensing of hepatic cDC1 by CD4⁺ T cells, and are thus associated with the generation of intrahepatic CD8⁺ CTLs and memory T cell differentiation. As the number of cognate T cells determines the effectiveness of the immune response, this amplification might be beneficial during immune responses in a large organ such as the liver, particularly against pathogens that spread quickly in hepatocytes such as HBV and HCV[39,40]. Similarly, immunization-induced generation of a large number of intrahepatic resident memory CD8⁺ T cells recognising a malarial antigen has been shown to be critical to protect recipient mice challenged with the malaria parasite;[41,42] amplification in portal tracts would also ensure that the number of memory CD8⁺ T cells generated in the liver is sufficiently large to provide effective immunity. While the results of this study suggest that CD40-CD40L interactions involving liver cDC1s play a dominant role in promoting the intrahepatic CD8⁺ T cells response, it is currently unknown whether CD40-CD40L interactions with other liver APC populations could play a role in promoting effective T cell immunity within the liver and this question remains the subject of ongoing investigation.

It is interesting to note that although portal tracts contain professional APCs and are able to facilitate the transfer of CD4 help to CD8⁺ T cells, they do not possess all the properties generally associated with SLOs. The main differences are not only structural (extended vs compact compartment), but also relate to anatomical and vascular differences that dictate distinct rules of T cell access and egress. Portal tracts form a compartment continuous with the space of Disse, the intercellular area separating hepatocytes from the fenestrated sinusoidal endothelium, which lacks a basement membrane[37,43]. This unique anatomy allows antigens derived from hepatocytes and the blood circulation to easily access portal tracts where they could be captured and cross-presented by resident cDC1s. However, the late accumulation of cognate CD8⁺ T cells in portal tracts observed in this work would suggest that it is antigen-experienced rather than naïve CD8⁺ T cells that encounter antigen in this compartment. Naïve T cells are too large to passively go through LSEC fenestrations and are unlikely to express the chemokine receptors and adhesion molecules that would allow efficient migration through LSECs[44,45]. Consequently,

while naïve T cells circulating in the blood continuously traffic through SLOs after trans-endothelial migration through HEVs or via the lymph circulation, their inability to cross the sinusoidal and portal endothelial barrier makes it unlikely that they access the space of Disse and portal tracts. As T cells undergo activation, they acquire the ability to cross endothelial barriers[44]. Thus, while SLOs can be considered as the main site of T cell priming that support both primary and secondary activation of naïve and central memory T cells respectively, portal tracts likely exclusively support re-activation of effector and memory T cells in the periphery, amplifying the CD8[+] T cell response by further expanding the pool of CD8[+] T cells that were initially activated in lymphoid tissues.

The demonstration of CD4 help-mediated CD8[+] T cell expansion in portal tracts and PCV regions provides insight into the significance of periportal and PCV infiltrates long observed in a range of liver diseases with an immune-mediated component, including viral hepatitis, autoimmune hepatitis, liver transplant rejection and metabolic associated fatty liver disease (MAFLD)[46–49]. Regardless of underlying etiology, immune-mediated inflammation in the liver is generally associated with lymphocyte aggregates detected predominantly in and surrounding portal tracts, as well as PCV regions. Particularly in HCV infection, these clusters can develop to resemble lymphoid tissues, with aggregates referred to as tertiary lymphoid structures. Such structures do not typically appear in healthy human liver tissue, but develop during chronic immune responses to liver antigens[46–49]. These observations are particularly intriguing in the case of viruses that target hepatocytes, but are paradoxically associated with lymphocyte accumulation in portal and PCV regions not targeted by the virus. While the size and composition of periportal infiltrates are routinely used to assess the severity of liver inflammation[47], their immunological significance, the underlying mechanisms leading to their formation, and their role in influencing the outcome of liver disease have not been resolved. Our findings hint that periportal lymphocyte aggregates reflect the robust expansion of T cells undergoing secondary activation in these areas after recognizing their antigen (cross)-presented by portal cDC1s. Considering that a strong anti-viral CD4[+] T cell response is a prerequisite for robust anti-viral CD8[+] T cell immunity and spontaneous viral clearance in individuals infected with HCV or HBV[8,12], it is likely that a significant role of the CD4[+] T cell response is to license hepatic cDC1s in order to expand the cognate anti-viral specific CTL response in portal tracts and increase the chances of controlling and clearing the virus. As the current work used a non-replicative vector to transduce a defined population of hepatocytes, the model does not completely recapitulate the setting of acute hepatotropic infection. However, future development of replicative vectors allowing similar analysis of CD8[+] and CD4[+] T cell responses to de novo hepatocyte-expressed antigen should allow further exploration of these hypotheses.

Although the molecular signals required to recruit cDC1s, CD8[+] T cells and CD4[+] T cells into portal tracts remain incompletely understood, it is possible to speculate based on current literature. The liver generates large amounts of lymph fluid, approximately 75% of which flows from the sinusoids into the space of Disse, and drains into the portal tract interstitium where it is collected by the portal lymphatic vasculature[37]. Thus, the migration of intrahepatic T cells and cDC1s to portal tracts might be influenced by the large flow of lymph fluid. It is also possible that chemokine, and/or adhesion molecule receptor-ligand interactions play an important role in this process, a notion supported by recent reports demonstrating chemokine- and adhesion molecule-dependent positioning of liver resident immune cells including KCs, CD8[+] T$_{RMS}$ and NKT cells[50,51]. It has been suggested that TCR recognition of peptide:MHCI on hepatocytes facilitates CD8[+] T cell extravasation from the sinusoidal lumen into the space of Disse[52,53], and this likely enhances CD8[+] T cell migration to portal tracts in our model. It is currently unknown whether TCR recognition of

peptide:MHCII enhances the extravasation of intrahepatic CD4[+] T cells from the sinusoids, however, it is possible that hepatocyte-expressed antigens presented by KCs, LSECs, cDCs or monocyte-derived macrophages play an important role[24,25,54,55].

While our results suggest that licensed hepatic cDC1s promote the CD8[+] T cell response in the liver via upregulation of co-stimulatory molecules, the precise molecular and cellular mechanisms that promote an increased effector and memory CD8[+] T cell pool remain incompletely understood. Future work addressing the underlying mechanisms that regulate DC and T cell trafficking in the liver, as well as the mechanisms promoting intrahepatic CD8[+] T cell proliferation, migration, and survival, could potentially lead to the development of therapeutic strategies to promote beneficial outcomes in liver disease. Targeting liver antigens to portal tracts and promoting their (cross)-presentation by cDC1s might be an effective strategy to improve the efficacy of vaccines against liver tropic pathogens or tumours. Strategies aimed at manipulating CD40 signalling in liver cDCs could potentially be utilized to modulate CD8[+] T cell expansion within portal tracts and T cell immunity in the liver.

In conclusion, these findings indicate that: (i) CD4 help via cognate DC licensing can be delivered to CD8[+] T cells outside of secondary lymphoid organs (SLOs); (ii) portal tracts and PCV regions represent distinct immunological niches within the liver, specialized in facilitating the T cell - cDC1 interactions necessary for the transfer of CD4 help to intrahepatic CD8[+] T cells; and (iii) cognate licensing of cDC1s in portal tracts and PCV regions amplifies CD8[+] T cell immunity in the liver.

## Methods
All experimental procedures involving mice were carried out according to protocols approved by the Sydney Local Health District Animal Welfare Ethics Committee (Royal Prince Alfred Hospital, Sydney, Australia) and were performed according to relevant national guidelines for husbandry and experimentation on laboratory animals. (Protocols 2020-005, 2021-013, 2016-006, 2016-041).

### Mice
All mice were imported, bred and housed in the animal facilities of the Centenary Institute.

C57BL/6 mice were obtained from Animal BioResources (ABR) (Moss Vale, NSW, Australia). 6-12 weeks old C57BL/6 mice were used for all experimentation. gBT-1 T cell receptor (TCR) transgenic mice, expressing a TCR specific for gB$_{498-505}$ (SSIEFARL) bound to H2-Kb[19] were crossed with C57BL/6.RAG1$^{-/-}$ mice[56] to generate gBT-1.RAG1$^{-/-}$ mice. gBT-I.RAG1$^{-/-}$ mice were crossed with C57BL/6.CD45.1 mice (B6.SJL-Ptpra$^a$ Pepc$^b$/BoyJ), imported from the Animal Resources Centre facility (WA, Australia), or mT/mG mice (Gt(ROSA)26Sor$^{tm4(ACTB-tdTomato,-EGFP)Luo}$)[57], kindly provided by Prof. Wolfgang Weninger, to generate gBT-1.RAG1$^{-/-}$.CD45.1 or gBT-1.RAG1$^{-/-}$.tdTomato mice, respectively. P25 TCR transgenic mice, expressing a TCR specific for the immunodominant peptide P25$_{240-254}$ (FQDAYNAAGGHNAVF) of *Mycobacterium tuberculosis* antigen 85b (Ag85b) bound to I-Ab[20], were crossed with C57BL/6.RAG1$^{-/-}$ mice to generate P25.RAG1$^{-/-}$ mice. P25.RAG1$^{-/-}$ mice were crossed with C57BL/6.CD45.1 mice or C57BL/6.uGFP mice[58] to generate P25.RAG1$^{-/-}$.CD45.1 or P25.RAG1$^{-/-}$.GFP mice, respectively. P25 mice were kindly provided by Prof Jamie Triccas with permission from Prof. Kiyoshi Takatsu. XCR1-Venus and XCR1-DTRvenus mice[26] were maintained in house at the Centenary Institute animal facility.

### Recombinant adeno-associated viral (rAAV) vectors
The rAAV generated for this study, designated rAAV$^{gB-Ag85b}$, is based on vectors described in our previous studies[21,22]. Briefly, rAAV2 vectors expressing genes of interest under the control of ApoE/hAAT promoter/enhancer elements were packaged into the hepatotropic type 8 serotype capsid, resulting in hepatocyte specific expression of the

gene of interest. rAAV plasmids were designed and constructed in the host lab and packaged by the Vector and Genome Engineering Facility (Children's Medical Research Institute, Westmead, NSW, Australia). rAAV.GFP.F2A.gB-Ag85b (rAAV$^{gB-Ag85b}$) is bicistronic and in addition to gB-Ag85b, encodes for green fluorescent protein (GFP). gB-Ag85b is a fusion protein containing a truncated membrane-bound ovalbumin (mOVA), in which the wild type OVA$_{257-264}$ (SIINFEKL) epitope has been replaced by gB$_{498-505}$ (SSIEFARL), and Antigen 85b (Ag85b) from *Mycobacterium tuberculosis*, containing the P25$_{240-254}$ (FQDAY-NAAGGHNAVF) epitope.

## Adoptive T cell transfer

Single cell suspensions of pooled lymph node cells from gBT-1 (RAG1$^{-/-}$.CD45.1 or RAG$^{-/-}$.tdTomato) and P25 (RAG1$^{-/-}$.CD45.1 or RAG$^{-/-}$.GFP) TCR Tg mice were made by pressing the tissue through an 80-gauge mesh sieve and washing twice with RPMI 1640 supplemented with 2% FCS. In some experiments cells were labelled with CSFE or CellTrace Violet (CTV; Thermo Fisher Scientific) according to the manufacturer's protocols. Lymph node cells were counted and then resuspended in sterile PBS and injected i.v. into the lateral tail veins of recipient mice.

## In vivo treatments

Recipient mice received $3 \times 10^9$ viral genome copies of rAAV$^{gB-Ag85b}$ in sterile Dulbecco's PBS via intravenous injection in order to induce de novo expression of gB-Ag85b in 1% of hepatocytes. To achieve effective in vivo CD4$^+$ T cell depletion before initiating a CD8$^+$ T cell response, recipient mice were injected i.v. with 400 μg GK1.5 anti-mouse CD4 depleting mAb or Rat IgG isotype control (BioXcell, West Lebanon, USA) at day 3 and day 1 before treatment with rAAV (Fig. S1G–I). For CD4$^+$ T cell depletion beginning at days 3, 6 or 9 post-rAAV treatment, mice received 500 μg GK1.5 or isotype control on the specified time points. To maintain complete CD4$^+$ T cell depletion, recipient mice received 400 μg twice weekly for the first 14-days and once per week between 14-days post-rAAV treatment and animal sacrifice, as dictated by the time course of the experiment. In experiments assessing the effector function of liver memory CD8$^+$ T cells, mice were injected i.v. with 50 μg ARTC2.2 blocking s + 16a nanobody (Biolegend, San Diego, USA) 30 min before sacrifice to prevent potential death of memory CD8$^+$ T cells caused by ARTC2 mediated ADP-ribosylation and activation of the cytolytic P2X7 ion channel[41,59]. To block CD40L in vivo, mice were injected i.v. with 500 μg MR1 anti-mouse CD40L blocking mAb (BioXcell, West Lebanon, USA) or polyclonal Armenian hamster IgG isotype control (BioXcell, West Lebanon, USA) in PBS at the indicated time points. To stimulate the CD40 pathway in vivo, mice were injected i.v. with 100 μg MR1 anti-mouse CD40 agonist mAb or Rat IgG isotype control (BioXcell, West Lebanon, USA) at the indicated time points. XCR1$^{DTR/wt}$ and XCR1$^{wt/wt}$ mice received an intravenous injection of 25 ng/g diphtheria toxin (DTX) unnicked from *Corynebacterium diphtheriae* (Sigma-Aldrich, St. Louis, MI, USA) on day 9 post-rAAV treatment and every two days following until animal sacrifice.

## Hepatocyte isolation

To isolate hepatocytes, the portal vein was transection to allow perfusate outflow and the liver was perfused in situ retrogradely via the IVC using a 4.5 mL/min flow by sequentially injecting the following solutions prewarmed at 37 °C: 25 mL of HBSS, followed by 25 mL HBSS with 0.5 mM EDTA, 25 mL HBSS, and finally 25 mL HBSS containing 1 mg/mL collagenase type IV (Sigma-Aldrich, St. Louis, MI, USA) and 5 mM CaCl$_2$[21]. After the final perfusion, gallbladders were excised, livers were removed *en bloc* and hepatocytes were gently teased from the liver into RPMI + 5% FCS where they were kept on ice for further analysis.

## Isolation of liver, lymphoid tissue, GI tract and blood lymphocytes

To isolate liver lymphocytes, livers were retrogradely perfused in situ with 5 ml of PBS via the IVC, following transection of the portal vein to allow outflow of perfusate[21]. Gallbladders were excised, and livers were then removed *en bloc*. Livers were gently pressed through an 80-gauge stainless steel mesh sieve in RPMI + 2% FCS, then centrifuged at 500 g for 5 min at 4 °C. The pellet was resuspended in a solution of 36% isotonic Percoll (GE Life Sciences, Parramatta, NSW, Australia) in PBS solution and centrifuged at 800 g for 12 min at 4 °C. The cell pellet was washed with RPMI + 2% FCS followed by red cell removal buffer (150 mM ammonium chloride, 0.1 mM EDTA disodium, 12 mM sodium hydrogen carbonate in triple distilled water), followed by a further wash in RPMI + 2% FCS. Lymphocytes were isolated from lymph nodes and spleens by pressing the tissue through an 80-gauge stainless steel sieve and washing in RPMI + 2% FCS.

To isolate GI Tract lymphocytes, the small intestine, large intestine, and Peyer's patch tissues were treated sequentially with PBS containing 1 mM DTT at room temperature for 10 min, twice with 5 mM EDTA at 37 °C for 10 min to remove epithelial cells, and then minced and dissociated in digestion buffer (RPMI) containing collagenase (1 mg/ml collagenase D; Roche), DNase I (200 μg/mL; Sigma) with constant stirring at 37 °C 55 min[60]. Leukocytes were collected at the interface of a 40–80% Percoll gradient (GE Healthcare). For blood lymphocytes, blood was collected by cardiac puncture into tubes containing Alsever's solution (Sigma-Aldrich, St. Louis, MI, USA). Peripheral blood mononuclear cells (PBMC) were prepared by lysing red blood cells with two washes of red cell removal buffer followed by washing with RPMI + 2% FCS. Single cell suspensions were stored in RPMI + 2% FCS on ice for further analysis.

## Isolation of liver DCs

Livers were retrogradely perfused in situ with 5 ml of 37 °C HBSS containing Ca and Mg, as well as 1 mg/mL collagenase type IV (Sigma-Aldrich, St. Louis, MI, USA) via the IVC, following transection of the portal vein to allow outflow of perfusate. Gallbladders were excised, and livers were then removed, chopped finely and incubated for 30 min in HBSS with Ca and Mg, containing 1 mg/mL collagenase type IV and 1 ug/mL DNase Type I (Roche, Basel, Switzerland) at 37 °C. Livers were gently pressed through an 80-gauge stainless steel mesh sieve in RPMI + 2% FCS, then centrifuged at 440 g for 5 min at 4 °C. The pellet was resuspended in a solution of 33% isotonic Percoll (GE Life Sciences, Parramatta, NSW, Australia) in PBS solution and centrifuged at 700 g for 12 min at room temperature with no brake. The cell pellet was washed with RPMI + 2% FCS followed by red cell removal buffer, followed by a further wash in RPMI + 2% FCS. Single cell suspensions were stored in RPMI + 2% FCS on ice for further analysis.

## Ex vivo peptide stimulation of T cells and intracellular cytokine staining

Liver and spleen lymphocyte cell suspensions were incubated in RPMI + 5% FCS containing anti-mouse CD107a mAb, Golgistop protein transport inhibitor (BD Biosciences, North Ryde, Australia) and 0.1 μg/mL peptide at 37ºC in 5% CO$_2$ for 4 h. Cells were washed in FACS buffer (2% FBS, 0.5 mM EDTA, 0.05% azide in PBS) and stained for surface markers and with ZombieUV viability dye (Biolegend, San diego, USA). Cells were washed with FACS buffer, then washed with PBS before fixing with 1% paraformaldehyde in PBS for 20 min at room temperature in the dark. Cells were washed twice with FACS buffer and incubated in FACS buffer containing 0.25% saponin (Sigma-Aldrich, St. Louis, MI, USA) and anti-mouse cytokine mAbs overnight at 4 °C in the dark. Cells were washed three times and resuspended in FACS buffer for flow cytometric analysis.

## Antibody staining, flow cytometry and cell sorting

Antibodies used for staining are provided in Supplementary Table 1. Single cell suspensions were incubated with 2.4G2 antibody for 20 min to block FcγII receptor-mediated binding, before adding antibodies diluted in FACS buffer and incubating on ice for 30 min. Stained cells were passed through a filter and, where appropriate, DAPI was added to a final concentration of 0.1 mg/ml immediately before acquisition to exclude dead cells from analysis. Flow cytometric cell counting was performed using AccuCount Blank beads (Sphero, Chicago, USA). 100 ul of cell suspension was taken from each sample and 10,000 beads were added in 100 ul FACS buffer. Cells were passed through a filter and DAPI was added to a final concentration of 0.1 mg/ml immediately before acquisition. For intracellular staining of Ki67, cells were incubated with antibodies diluted in FACS buffer cells and ZombieUV cell viability dye (Biolegend, San diego, USA) for 30 mins on ice. Cells were washed with FACS buffer, then washed with PBS before fixing with 1% paraformaldehyde in PBS for 20 min at room temperature in the dark. Cells were washed twice with FACS buffer and incubated in FACS buffer containing 0.25% saponin and anti-mouse Ki67 (SolA15) for 2 h at 4 °C in the dark. Flow cytometric analysis was performed using an LSR-Fortessa flow cytometer (Becton Dickinson, North Ryde, NSW, Australia), with data acquisition on a windows computer using FACSdiva (Becton Dickinson). Analysis was performed using Flowjo 10 software (TreeStar Inc) on a Macintosh computer (Apple, Cupertino, CA, USA). Cell sorting was performed using a BD Influx 7 laser cell sorter (Becton Dickinson, North Ryde, NSW, Australia).

## DC – T cell co-culture

cDC1s were isolated from the livers of mice treated with rAAV$^{gB-Ag85b}$ or PBS 12-days prior. Enrichment for DCs was performed using MACS CD11c-positive selection kit (Miltenyi), before cDC1s were sorted based on CD11c, MHCII, XCR1 and CD11b expression (CD11c⁺MHCII⁺XCR1⁺CD11b⁻). Naïve TCR Tg T cells were isolated from the LNs of gBT-1 or P25 TCR Tg mice and labelled with CTV following the manufacturers protocol (Thermo Fisher). 10,000 sorted cDC1s were co-cultured with 100,000 CTV-labelled gBT-1 or P25 TCR Tg T cells in sterile RPMI containing 10% FCS, 50 μM β-mercaptoethanol (Sigma), Glutamax (Gibco) and 1% penicillin/streptomycin (Gibco). Co-culture was incubated at 37 °C, 5% $CO_2$ and 20% $O_2$ for 72 h before T cell activation and proliferation were assessed by flow cytometry.

## Preparation of tissue for immunofluorescence microscopy

Livers were retrogradely perfused in situ with 5 ml of cold PBS, then 15 ml of cold 2% PFA in PBS, following transection of the portal vein to allow outflow of perfusate. Gallbladders were excised, and livers were then removed and placed into 20 mL of 2% PFA and incubated overnight or 8 h at 4 °C in the dark. Livers were washed with PBS and embedded in 2% agarose. 100–200 um sections were cut using a vibratome (Leica VT1200) and placed in blocking buffer containing 4% BSA, 5% goat or donkey serum, 5% mouse serum and 0.3% Triton-X 100 in dPBS overnight at 4 °C in the dark with gentle agitation. Tissue sections were stained with fluorophore-conjugated primary antibodies in blocking buffer overnight at 4 °C in the dark with gentle agitation then washed 3 times in Dulbecco's PBS with 0.1% Triton-X 100. If unconjugated purified antibodies were used, washed tissue sections were stained with the appropriate Alexafluor-conjugated anti-species antibody overnight at 4 °C in the dark with gentle agitation. Tissue sections were then washed 3 times in Dulbecco's PBS with 0.1% Triton-X 100 and when required incubated with 1 ug/mL DAPI for 1 h at 4 °C in the dark, then washed 3 times. For analysis of large volumes of tissue, liver sections were cleared and mounted using Ce3D clearing solution[61]. Alternatively, tissue samples were mounted with Dabco antifade mounting medium (Sigma-Aldrich, St. Louis, MI, USA).

## Confocal microscopy and imaging analysis

Images were acquired using a Leica DMi8 SP8 inverted confocal microscope with a motorized stage for tiled imaging (Leica Microsystems, Germany). Hybrid Detectors were used for all acquired images. Images showing large liver areas were generated by stitching multiple tiles following acquisition using the stitching function of the Leica Application Suite X (LAS X). All images were processed and analysed using Imaris software version 9.6 (Bitplane, Oxford Instruments, UK). T cell cluster and density quantification in different liver compartments was performed manually using Imaris 9.6. Clusters of proliferating gBT-1 T cells were defined as ≥4 Ki67⁺ gBT-1 T cells in direct contact. Portal tracts were identified through staining for LYVE-1$^{high}$ lymphatic endothelial cells, peri-central venous regions were identified at vascular structures absent of LYVE-1$^{high}$ lymphatic endothelial cells and parenchymal/sinusoidal regions were identified as regions of DAPI⁺ hepatocytes without portal tracts or central veins. For nearest neighbour analysis, CD45.1⁺ and tdTomato⁺ gBT-1 T cells were segmented in portal tracts, PCV regions and the sinusoidal compartment using the surface function in Imaris 9.6. Positional data of segmented cells was exported and used to determine the fraction of nearest non-self-neighbours of CD45.1⁺ gBT-1 T cells. For CD4⁺- CD8⁺ T cell and T cell-XCR1⁺ cDC1 proximity analysis, CD45.1⁺ gBT-1 T cells, GFP⁺ P25 T cells and Venus⁺ XCR1 cells were segmented in Imaris 9.6 using the surface function and the average distance between cells was calculated in portal tracts, PCV regions and the sinusoidal compartment using the XTension plugin "Spots and Surfaces Distance" MATLAB version: 9.13.0 (The MathWorks Inc. 2022. Natick, Massachusetts: The MathWorks Inc.). To quantify the level of expression of CD40 by liver cDC1s in situ, image analysis was performed using the surface function of Imaris software to segment cells expressing both MHCII and CD103, and the mean fluorescence intensity of CD40 fluorescence for each cell was exported and plotted. Liver tissue volume and area were calculated manually using the surface creation function in Imaris 9.6.

## Statistical analysis

Data was plotted and statistics were determined using Prizm v9 software (Graphpad software Inc, San Diego, California, USA). Unless otherwise stated, data is represented as mean ± standard error of the mean (SEM) and is representative of at least 2 independent experiments. All images are representative of at least $n = 3$ mice and at least 2 independent experiments. Statistical analysis was performed using Students $t$ test and one-way ANOVA, or Mann–Whitney $U$ test and Kruskal–Wallis test, for normal and non-normal distributed data respectively. All statistical tests were two-tailed and unpaired. A $p$-value less than 0.05 ($p < 0.05$) was used to assess statistical significance.

## Reporting summary

Further information on research design is available in the Nature Portfolio Reporting Summary linked to this article.

# Data availability

Source data are provided with this paper. The data that support the findings of this study are available within the article, supplementary files, source data files or from the corresponding authors on request. Source data are provided with this paper.

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

## Acknowledgements

We thank the Centenary Institute Bioresources Facility, the Centenary Imaging Facility and the Sydney Cytometry Core Research Facility (a joint initiative of Centenary Institute and the University of Sydney) for their excellent technical support. We particularly thank Dr Angela Fontaine for sharing her imaging expertise. This work was supported by an NHMRC Australia Project Grant (APP APP1146677). KE was the recipient of a PhD Scholarship from the University of Sydney.

## Author contributions

Conceptualization: D.G.B., P.B. and K.E.; Methodology: K.E., P.B., D.G.B., J.M.M.K. and L.L.; Validation: K.E., R.K., C.M. and L.E.H.; Formal Analysis: K.E., R.K., L.E.H., C.M. and D.K.; Investigation: K.E., R.K., L.E.H., C.M. and J.M.M.K.; Resources: T.K., W.R.H., L.L. and M.B.; Writing (first draft): K.E., D.G.B. and P.B.; Review & Editing: K.E., P.B., D.G.B., G.W.M. and W.R.H.; Visualization: K.E. and D.K.; Supervision: P.B., D.G.B., and G.W.M.; Project Administration: P.B. and D.G.B.; Funding Acquisition: P.B., D.G.B., and G.W.M.

## Competing interests

The authors declare no competing interests.

## Additional information

**Supplementary information** The online version contains Supplementary Material available at https://doi.org/10.1038/s41467-024-45612-5.

