## [Peer Review File · Nature Communications]

REVIEWER COMMENTS

Reviewer #1 (Remarks to the Author):

In this study, Dr. Kieran English et al discovered a unique extra-lymphoid structure in pericentral venous (PCV) regions, constructed by effector CD4+ T cells, memory CD8+ T cells, and antigen-presenting cDC1 cells. The authors also found that effector CD4+ T cells licensed cDC1 cells locally, leading to antigen presentation by cDC1 cells in the local area, and expansion of antigen-specific CD8+ T cells. These findings are both interesting and significant.

However, there are still some questions that the authors need to address. First, they should compare this structure with third lymphatic structures (TLO) in the liver, as there are existing TLOs in this organ. Second, the authors need to detect the ratio of co-transferred naive gBT-1 and P25 T cells in draining lymph nodes (dLN) to exclude the effect of dLN on these two cell populations' proliferation, or block the lymphatic tissue using FTY720. Third, to confirm that CD4+ T cells are essential for licensing cDC1, the authors should deplete CD4 T cells, use blocking or neutralizing antibodies of CD40/CD40L. Finally, to confirm the importance of the interaction of CD40-CD40L, the authors should compare depletion of cDC1 with inducible knock-out CD40 of cDC1.

In addition to these issues, there are still some questions that need further discussion. For example, it is unclear whether CD4+ T cells are activated in situ or migrate from dLN, and the characteristics of these CD4+ T cells. Additionally, the authors observed significant differences in CD8+ T cell memory stages, but they did not show differences in other results. It would be helpful for the authors to provide additional information on this point.

Overall, this study presents a fascinating and beautiful discovery, with important implications for our understanding of lymphoid structure and cellular interactions. However, the authors should address these concerns and provide additional information to strengthen their findings.

Reviewer #2 (Remarks to the Author):

- "What are the noteworthy results?"

This manuscript describes the role of CD4+ T cells in "helping" CD8+ T cell-mediated intrahepatic immunity to liver-expressed antigens. By using a mouse model, the authors show that CD4+ T cells

license cDC1 cells via a CD40-CD40L-dependent axis directly in the liver, resulting in a local expansion of CD8+ T cells. The authors identify intrahepatic perivascular compartments as the site where these local interactions occur.

- "Will the work be of significance to the field and related fields? How does it compare to the established literature? If the work is not original, please provide relevant references."

The novelty of the study is the detailed analysis of a key immunological mechanism, namely CD4 help.

The de novo expression of a GFP-expressing antigen directly on hepatocytes coupled to the transfer of ag-specific T cells is laudable, since it allows the authors to visualize location and possible interactions of ag-specific T cells and APCs locally in the liver.

We suggest that the authors clarify the first sentence of the abstract to explicitly describe what the novelty of this manuscript is.

- "Does the work support the conclusions and claims, or is additional evidence needed?"

Are there any flaws in the data analysis, interpretation and conclusions? - Do these prohibit publication or require revision?

Is the methodology sound? Does the work meet the expected standards in your field?

Is there enough detail provided in the methods for the work to be reproduced?"

Overall, the experiments are well designed and meet the expected standard in our field.

However, some of the claims presented in the manuscript seem not to be substantiated by the data presented and some controls. Specifically, experiments involving injection of neutralizing antibodies are missing.

Major points:

- Fig.1 and S1: We suggest that the authors show that CD4 depletion was achieved, while the bulk of CD8 was not affected by it (Fig.1F, for example).

In addition, it would be important to specify if the pooled LNs are the draining LN of the liver or whether others are also included. Finally, it would be important to include an additional organ, e.g., the intestine, as further control for the specificity (see below).

- Fig.2:

- We suggest adding the gates and the % on the dotplots reported (A).

- As in Fig.1, we recommend showing the individual values throughout all the panels.

- It might be important to consider that Ki67 is expressed by any cell that is not in G0, in other words, also by cells that are not actively proliferating and have just entered the cell cycle, e.g. cells in G1) (Gerdes et al. JI, 1984. PMID: 6206131. Therefore, the evaluation of proliferation of gBT-I CD8 T.

cells should be carried out via 2n/4n DNA content staining (Fig.2G-H and S2B-C). This way, cells in G1 can be discriminated by actively proliferating cells in G2/M/S.

· Fig.3:

-The approach used by the authors to discriminate between local expansion and accumulation of recirculating dividing cells is smart, but it might not be sufficient. How do the authors exclude the possibility that the cells expand in the dLN (or despite being unlikely, in other organs)? One way to test this would be to use FTY720 soon after priming and before expansion. In addition, in vivo labeling approaches after priming, but before expansion, would help to suggest liver-residency of gBT-I cells. If none of these is possible, we suggest to tone down these conclusions.

Finally, the results could be biased by some underlying defect in expansion of tdTomato+ gBT-I cells, as opposed to CD45.1+ counterparts. The opposite ratio (9.000 CD45.1 +1.000 dtTomato) needs to also be tested.

-The quantification of the data shown in E and F is mandatory to support the conclusions.

· Fig.4:

- Please specify what the Y axis indicates.

- Statistical testing needs to be performed in Fig.4A to support the claim that “the number of gBT-1 T cells in the liver following activation in the absence of CD4+ T cells from days 3, 6 or 9, were statistically indistinguishable from the number of gBT-1 Tg CD8 T cells present following activation in the absence of CD4+ T cells from day 0”.

-The conclusion of the authors for Fig.4 is speculative. Confocal images (Fig.G-I) do not show direct cell-cell interactions, but rather proximity of some gBT-I cells to P25 CD4+ T cells. In order to claim that there is a physical, direct cell-cell interaction between CD8 and CD4 T cells within PCVs, the merge between the two colors representing CD8 and CD4 needs to be visible. We only see proximity.

-The quantification of the interactions (G-I) is needed.

· Fig.5:

-The quantification is needed for panel D.

-We suggest that the authors test if CD40-CD40L axis was successfully blocked when injecting anti-CD40L.

- Statistical assessment in panel A, between CD4-depleted and anti-CD40L-treated groups needs to be performed, in order to support the respective statement.

-If possible, it would strengthen the results to show that by blocking CD40-CD40L axis the interaction between the cells are reduced.

· Fig.5-6:

-The authors conclude that CD4, CD8 and APC cells are in direct (and cognate) contact. This is however not shown by the confocal imaging, as no merge between the membranes of the cells is visible. As above, the cells seem to be in close proximity, but they do not appear to be in a direct and cognate interaction, as it should be the case in an immunological synapse-like setting as the one described by the authors.

- Authors should show original confocal images for the quantifications shown in graphs (Fig.6F-I)

-Following depletion of XCR1+ cells, gBT-I cells are markedly decreased. Does the depletion of XCR1+ cells affect their viability? Or do gBT-I cells fail to migrate to the liver? Does XCR1+ cell depletion affect the bulk of CD8 memory pool? The authors should experimentally address, or at least discuss, these options.

Discussion:

· Several references are missing from the discussion, e.g. "... challenge the paradigm that cognate DC licensing is restricted to SLOs"; "The portal interstitium surrounds the branches of the portal vein, hepatic artery and biliary tree that extend throughout the entire liver; thus, it occupies a significant volume and represents a vast compartment that extends throughout this large solid organ". All the last 3 paragraphs are almost not referenced at all.

Reviewer #3 (Remarks to the Author):

The authors have developed elaborate methods that combine two antigens (gB and Ag85b) with known TCRs of CD8 (gBT-I) and CD4 (P25) cells in order to determine the role of CD4+ cells in the activation of CD8+ T cells in the liver. Using these methods, the authors showed that the expansion of antigen-specific CD8 T cells in the liver of gB-Ag85b-treated mice is dependent on CD4+ T cell help and APC support through CD40-CD40L interaction. The manuscript and experiments are well organised and the subject is of interest to both immunologists and liver researchers. However, several points need clarification.

Fig.1:

Anti-CD4 treatment depletes CD4+ T cells and possibly MAIT cells; reconstitution experiments using gBT-I CD8 cells together with CD4 T cells or MAIT cells would clarify this issue. The authors also need to show that “intrahepatic CD8+ cells” are not MAIT cells (MAIT cells include a CD4 CD8 double positive cell population).

Dynamics of CD8 activation (CD25, 69, IFN γ , CD107a) and proliferation (Ki67) is not clear in the Figure 1 and 2. The authors should indicate the percentage and number at each time point (D0-D46). This will clarify why the authors chose Day 9, 12, and 15 for representative time point in the figure 2 and after.

Fig.3D-F:

The interpretation of these panels is perplexing. The representative panels shown in Figures 3E and F would seem to indicate a predominance of CD45.1+ tdT- CD8 cells in both PV and CV regions. However, if CD45.1+ tdT- and CD45.1- tdT+ CD8 cells have identical or similar characteristics, the ratio of CD45.1+ tdT- to CD45.1- tdT+ CD8 cells should remain 1:9 across the liver. The authors show low magnification images to help the reader understand the frequency of these cell clusters in the PV and CV regions. The ratios of CD45.1+ tdT- and CD45.1- tdT+ CD8 cells should also be summarised for all ROIs in the liver. Flow cytometric analysis of CD8 T cells throughout the liver would also be helpful.

CD40 is expressed on a variety of APCs, including Kuepfer cells, B cells and hepatic stellate cells. The authors should deplete these APCs and clarify why the CD40-40L interaction between T and cDC1 is important and not other APCs.

What types of CD4+ T cells are critical in activation and proliferation of hepatic CD8+ cells in this system? Fig. 4 shows that almost a week is required to detect CD4+ T cells originated from adoptively transferred naïve CD4+ T cells. However, it is possible that both rapidly activated and differentiated cell and activated tissue resident cd4+ T cells can enhance this process. The authors should investigate the kinetics of CD4 and CD8 in the mice reconstituted with mixed bone marrow of gBT-I and P25 mice and treated with the AAV.

Minor comments:

Untransferred control data enables readers to better understand antigen specificity in proliferation and activation of CD8+ cells in the liver.

AAV is infected in relatively small portion of cells. To eliminate the possibility of non-specific activation of hepatic CD8+ T cells, the author should show the production of IFN γ and CD107a of host CD45.2 CD8+ T

cells under CD4-full or –depleted conditions. In addition, the proportion of IFN γ + and CD107a+ CD8 cells seems comparable with or without anti-CD4 treatment. The changes in the numbers and proportions of CD8, IFN γ /CD107a CD8 cells in the liver and spleen from both host and donor cells are necessary. It is helpful for readers to understand the whole picture of this system.

The authors show the image of lower magnifications for providing the frequencies of T cell accumulation in the whole liver in Fig.3, 4, 5, and 6.

GFP+ hepatocytes are attacked and eliminated by activated gBT-I CD8 T cells? The kinetics of GFP+ cells is also an interesting outcome of activated and proliferated CD8 T cells.

In typical cases, the liver diseases are categorised by the pathological dominancy, particularly periportal or pericentral dominancy, but this model affects both zone 1 and 3. The levels of AST and ALT may help understanding the main focus of inflammation in the liver. In addition, the authors should discuss what types of liver damage or what liver diseases this model represents.

Reviewer 1:

Comment 1: *First, they should compare this structure with third lymphatic structures (TLO) in the liver, as there are existing TLOs in this organ.*

Response: Tertiary lymphoid structures are observed in the setting of liver disease, as opposed to the immune networks observed within portal tracts that are present in the steady state, which from our data appear to function early in the response to hepatically-expressed antigen. We have added discussion of hepatic tertiary lymphoid structures to the manuscript (*p 18 of revised manuscript, lines 492-495*).

Comment 2: *Second, the authors need to detect the ratio of co-transferred naive gBT-1 and P25 T cells in draining lymph nodes (dLN) to exclude the effect of dLN on these two cell populations' proliferation, or block the lymphatic tissue using FTY720.*

Response: We thank the reviewer for this comment, and agree that the contribution of the liver draining LNs to the help-dependent expansion of CD8⁺ T cells in portal tracts and PCV regions of the liver is an important question. We have therefore performed a series of experiments involving FTY720 treatment of mice that have received gBT-1 T cells and rAAV^{gB-Ag85b} inoculation. FTY720 treatment beginning at 2 days post-rAAV administration led to a significant reduction in the total number of intrahepatic gBT-1 T cells and Ki67⁺ gBT-1 T cell numbers in portal tract and pericentral venous clusters at 12-day post rAAV (**Updated Figures 6 A-D and S6A, B, D**). This decrease is consistent with a requirement for initial T cell activation in lymph nodes for later gBT-1 T cell expansion within the liver. This data is congruent with our previous results suggesting that the primary sites of activation of gBT-1 and P25 T cells following antigen expression by hepatocytes are the liver draining LNs (*Tay, S et al. PNAS 2014, 111: E2540*). In contrast, treatment of mice with FTY720 beginning at 9 days post rAAV administration did not significantly alter the total number of intrahepatic gBT-1 T cells or the number of Ki67⁺ gBT-1 T cells in portal tract and pericentral venous clusters at 12-days post rAAV (**Updated Figures 6 E-H and S6 C, E**). These findings confirm that at this later time point, when our CD4⁺ T cell depletion and CD40L blockade experiments indicated that CD4 help was being mediated, a further contribution of T cells from lymphoid tissues was not required for the expansion of gBT-1 T cells within PTs and PCV regions. When combined, the findings from these new experiments are consistent with CD4 help being delivered within the liver itself at this later time point. A new section has been added in the revised manuscript to describe these results (*p12-13, lines 311-340*).

Comment 3: *Third, to confirm that CD4+ T cells are essential for licensing cDC1, the authors should deplete CD4 T cells, use blocking or neutralizing antibodies of CD40/CD40L. Finally, to confirm the importance of the interaction of CD40-CD40L, the authors should compare depletion of cDC1 with inducible knock-out CD40 of cDC1.*

Response: The original version of the manuscript contains several experiments (outlined in **Figures 1C-1F; 2A-2J; 3A-3B; 4A-C and 7E**) in which we have depleted CD4⁺ T cells using GK1.5 anti-CD4 or isotype control monoclonal antibodies (mAbs), to assess the role of CD4 T cells in the CD8⁺ T cell response to rAAV^{gB-Ag85b} inoculation.

The role of CD40/CD40L interaction in the CD8⁺ T cell response to rAAV^{gB-Ag85b} inoculation was based on the following findings:

- Blocking CD40L-CD40 interactions using MR1 anti-CD40L mAbs, (**Figure 5A-5B**) after day 9 post rAAV inoculation leads to decreased numbers of transgenic CD8⁺ T cell in portal tracts in comparison to isotype control mAb treated mice.

- Providing CD40 signalling in the absence of CD4⁺ T cell help by treating CD4-depleted mice with anti-CD40 stimulating mAb (FGK45) overcomes the lack of endogenous CD4⁺ help (**Figure 5C**) and leads to increased CD8⁺ T cell expansion in the liver in comparison to isotype control mAb treated mice.

We believe these experiments demonstrate the requirement for CD40L-CD40 receptor ligand interactions for the intrahepatic expansion of CD8⁺ T cells recognising antigens expressed by hepatocytes.

Several recent publications have shown that CD40 expression on cDC1s is essential for the transfer of help via cognate DC licensing when antigens are cross-presented (*Hor et al. Immunity 2015, 43, 3:554-65; Eickhoff et al. Cell 2015, 162, 6:1322-37; Ferris et al. Nature 2020, 584, 7822:624-29*). Consistent with these findings, we have shown, in our model, that upregulation of CD40 expression on liver cDC1s is critically dependent on CD4 T cells. The key role of licensed cDC1s in the expansion of CD8 T cells in portal tracts was further demonstrated by the decreased CD8 T cell expansion following selective depletion of cDC1s after day 9 post rAAV inoculation (**Figures 7D-7I**).

We believe that our findings support a model in which CD40-expressing cDC1s play a dominant role in the CD4⁺ T cell mediated intrahepatic expansion of CD8⁺ T cells responding to antigens expressed by hepatocytes. The dominant role of CD40-expressing cDC1s in CD4 help to CD8 T cells is also consistent with the conclusions of other studies in different settings (listed above). While this is the most likely model explaining our findings, we cannot totally rule out the contribution of CD40-CD40L interactions involving other liver APCs in promoting the expansion of intrahepatic CD8⁺ T cells. We have mentioned this alternate possibility in the discussion of the revised manuscript (*p17, lines 457-461*).

Comment 4: *In addition to these issues, there are still some questions that need further discussion. For example, it is unclear whether CD4⁺ T cells are activated in situ or migrate from dLN, and the characteristics of these CD4⁺ T cells.*

Response: Findings presented in a previous study by our group indicate that CD4⁺ T cells responding to antigens expressed *de novo* by hepatocytes are most efficiently activated in the liver draining celiac and portal lymph nodes, with a small proportion activated in the liver (*Tay, S et al. J Immunology 2014 193: 2087*). These findings, together with our results described in Figure 4, indicate that CD4 T cells are primarily activated in the liver draining LNs and first migrate to the liver between 7 and 10 days post-rAAV mediated antigen expression by hepatocytes. We have amended the text to make this clearer to the reader (*p10, lines 264-273*)

While we agree that the characteristics of CD4⁺ T cells that respond to antigens expressed by hepatocytes represents an important question, we believe this is outside the scope of the current study.

Comment 5: *Additionally, the authors observed significant differences in CD8⁺ T cell memory stages, but they did not show differences in other results. It would be helpful for the authors to provide additional information on this point.*

Response: Significant differences in the number of functional intrahepatic CD8⁺ T cells were found in both memory stages (42 days post rAAV treatment) (**Figures 1C-1E**), and effector stages (15 days post rAAV treatment) (**Figures 2D, S2A-S2B**).

Reviewer 2:

Comment 1: *The novelty of the study is the detailed analysis of a key immunological mechanism, namely CD4 help. The de novo expression of a GFP-expressing antigen directly on hepatocytes coupled to the transfer of ag-specific T cells is laudable, since it allows the authors to visualize location and possible interactions of ag-specific T cells and APCs locally in the liver. We suggest that the authors clarify the first sentence of the abstract to explicitly describe what the novelty of this manuscript is.*

Response: We have amended the abstract to emphasize the novelty to the reader.

Comment 2: *Overall, the experiments are well designed and meet the expected standard in our field. However, some of the claims presented in the manuscript seem not to be substantiated by the data presented and some controls. Specifically, experiments involving injection of neutralizing antibodies are missing.*

Response: We assume that this comment applies to **Figures 5A-5B**, in which we assessed the role of CD40L/CD40 interactions in the CD8⁺ T cell response to rAAV^{gB-Ag85b} inoculation using neutralising monoclonal antibodies specific for CD40L (clone MR1) that block CD40L-CD40 interactions in vivo, or isotype control antibodies. In the revised manuscript, we have amended **Figures 5A-5B** to highlight the statistical significance or lack of significance between all groups, including those treated with isotype or neutralising antibodies.

Comment 3: *Fig.1 and S1: We suggest that the authors show that CD4 depletion was achieved, while the bulk of CD8 was not affected by it (Fig.1F, for example).*

In addition, it would be important to specify if the pooled LNs are the draining LN of the liver or whether others are also included.

Finally, it would be important to include an additional organ, e.g., the intestine, as further control for the specificity (see below).

Response: We have added plots and graphs to show that GK1.5 treatment resulted in effective CD4⁺ T cell depletion without demonstrable effect on the endogenous CD8⁺ T cell population (**Figures S1G-S1I**).

We have amended the text in the result section to explain exactly what we refer to as pooled LNs in Figure S1 (*p 5, line 111*).

Finally, we have performed experiments to confirm that the GI tract shows minimal evidence of CD8⁺ T cell proliferation following antigen expression by hepatocytes. These findings are shown in a new panel in Figure S2 (**Figure S2H**).

Comment 4: *Fig.2: We suggest adding the gates and the % on the dot plots reported (A). As in Fig.1, we recommend showing the individual values throughout all the panels.*

Response: We have ensured that all flow cytometry dot plots throughout the revised manuscript show gates and percentages. Where appropriate, all panels now show individual values.

Comment 5: *It might be important to consider that Ki67 is expressed by any cell that is not in G0, in other words, also by cells that are not actively proliferating and have just entered the cell cycle, e.g. cells in G1) (Gerdes et al. JI, 1984. PMID: 6206131. Therefore, the evaluation of proliferation of gBT-I CD8 T cells should be carried out via 2n/4n DNA content staining (Fig.2G-H and S2B-C). This way, cells in G1 can be discriminated by actively proliferating cells in G2/M/S.*

Response: We thank the reviewer for raising this important point. In response, we have carried out additional experiments to identify the proportion of gBT-1 T cells in G2/S/M phases of the cell cycle by performing Ki67 staining along with 2N/4N DNA content analysis (**Figures 2H-2I**). The results of these experiments suggest that the liver contains the highest proportion of CD8⁺ gBT-1 T cells in G2/S/M phases of the cell cycle at 12-days post rAAV treatment, when compared the liver draining lymph nodes, and that the presence of CD4⁺ T cells results in a significant increase in the proportion of CD8⁺ gBT-1 T cells in G2/S/M phases of the cell cycle in the liver and liver draining lymph nodes when compared to CD4⁺ T cell depletion (**Figure 2H-2I**).

Comment 6: *The approach used by the authors to discriminate between local expansion and accumulation of recirculating dividing cells is smart, but it might not be sufficient. How do the authors exclude the possibility that the cells expand in the dLN (or despite being unlikely, in other organs)? One way to test this would be to use FTY720 soon after priming and before expansion. In addition, in vivo labelling approaches after priming, but before expansion, would help to suggest liver-residency of gBT-1 cells. If none of these is possible, we suggest toning down these conclusions.*

Finally, the results could be biased by some underlying defect in expansion of tdTomato⁺ gBT-1 cells, as opposed to CD45.1⁺ counterparts. The opposite ratio (9.000 CD45.1⁺ +1.000 dtTomato) needs to also be tested.

The quantification of the data shown in E and F is mandatory to support the conclusions.

With regard to the reviewer's pertinent suggestion regarding experiments utilizing FTY720, please see response to Reviewer 1's comment 2 outlining the experiment undertaken in response to both reviewers' suggestions.

We have added a nearest neighbour analysis of CD45.1⁺ and tdTomato⁺ gBT-1 T cells in portal tracts, PCV regions and the sinusoidal compartment of the liver to both provide quantification and to assess biases caused by possible defects in tdTomato⁺ vs CD45.1⁺ gBT-1 T cells (**Figure 3H**). This analysis demonstrated that the ratio of cells in the sinusoids, as a whole, resembles the original adoptive transfer ratio. We have also added flow cytometric analysis of the ratio of tdTomato/CD45.1⁺ gBT-1 T cells in the inguinal lymph nodes (ILN) of mice that received an initial adoptive transfer of 1,000 CD45.1⁺ and 9000 tdTomato⁺ gBT-1 T cells, 15-days post rAAV treatment to show that the initial ratio is preserved following expansion and recirculation of gBT-1 T cells, suggesting negligible expansion bias (**Figure S3E**).

The results section has been amended to describe these new results (*p 8-9, lines 207-224*)

Comment 7: *Fig.4: Please specify what the Y axis indicates. Statistical testing needs to be performed in Fig.4A to support the claim that "the number of gBT-1 T cells in the liver following activation in the absence of CD4⁺ T cells from days 3, 6 or 9, were statistically indistinguishable from the number of gBT-1 Tg CD8 T cells present following activation in the absence of CD4⁺ T cells from day 0".*

The conclusion of the authors for Fig.4 is speculative. Confocal images (Fig.G-I) do not show direct cell-cell interactions, but rather proximity of some gBT-1 cells to P25 CD4⁺ T cells. In order to claim that there is a physical, direct cell-cell interaction between CD8 and CD4 T cells within PCVs, the merge between the two colors representing CD8 and CD4 needs to be visible. We only see proximity.

The quantification of the interactions (G-I) is needed.

Response: The axis label and indicators of non-significance have been added to the figure (**Figure 4A**).

We agree that Figure 4 does not provide convincing evidence of direct cell-cell interactions. We have toned down the conclusion to suggest proximity rather than cell-cell contact. The results section has been amended accordingly (*p10, lines 261-262*).

To demonstrate that CD8⁺ and CD4⁺ T cells are in closer proximity in portal tracts and central vein regions than in the sinusoids, we have quantified the average distance between gBT-1 and P25 T cells in portal tract, PCV and sinusoidal compartments of the liver at 12-days post rAAV. This quantification is illustrated in a new panel in Fig. S4 (**Figure S4F**) and described in the result section (*p10, lines 264-266*).

Comment 8: *Fig.5: The quantification is needed for panel D.*

We suggest that the authors test if CD40-CD40L axis was successfully blocked when injecting anti-CD40L.

Statistical assessment in panel A, between CD4-depleted and anti-CD40L-treated groups needs to be performed, in order to support the respective statement.

If possible, it would strengthen the results to show that by blocking CD40-CD40L axis the interaction between the cells are reduced.

Response: We have added quantification for panel D (**Figure S5B and S5C**).

We agree that confirming the successful blocking of the CD40-CD40L axis would be critical if we derived a negative result from our experiments. However, we believe that our results showing a significant decrease in the expansion of CD8⁺ T cells when treating mice with CD40L blocking mAbs show that the axis was successfully blocked in our experiments. These results are consistent with validation by the manufacturer (BioXcell) and the results of previous publications using the product.

Statistical assessment has been performed for panel A and added to **Figure 5A**.

While we do not expect that CD40-CD40L binding alone is responsible for inducing interactions between T cells, we believe that the downstream effects of CD40L-CD40 binding, such as chemokine/cytokine production by DCs could increase migration of T cells into portal tracts and interaction with DCs. We submit that this interesting question is outside the scope of the current study and is currently the subject of ongoing investigation.

Comment 9: *Fig.5-6: The authors conclude that CD4, CD8 and APC cells are in direct (and cognate) contact. This is however not shown by the confocal imaging, as no merge between the membranes of the cells is visible. As above, the cells seem to be in close proximity, but they do not appear to be in a direct and cognate interaction, as it should be the case in an immunological synapse-like setting as the one described by the authors.*

Authors should show original confocal images for the quantifications shown in graphs (Fig.6F-I)

Following depletion of XCR1+ cells, gBT-1 cells are markedly decreased. Does the depletion of XCR1+ cells affect their viability? Or do gBT-1 cells fail to migrate to the liver? Does XCR1+ cell depletion affect the bulk of CD8 memory pool? The authors should experimentally address, or at least discuss, these options.

Response: We have added a representative high-power image of a portal tract cluster in which gBT-1 T cells, P25 T cells are in direct cell-cell contact with the same XCR1⁺ cell, as shown by merging of the cell membranes, highlighted by staining with an antibody against CD45 (**Figure 7C**). Additional representative confocal images have also been added and shown in **Figure S7B**. Quantification of the average distance between gBT-1 T cells and XCR1⁺ cells, and P25 T cells and XCR1⁺ cells, in portal tracts, PCV regions and the sinusoidal compartment is shown in a new **Figure S7C**.

Our data suggest that while the total number of Ki67⁺ gBT-1 CD8⁺ T cells in portal and central vein clusters, as well as total numbers gBT-1 T cells in the liver, are significantly lower after the depletion of XCR1⁺ cDC1s, gBT-1 T cells are able to migrate to the liver (**Figure 7F-7H; Figure S7F**). However, we agree that we cannot rule out defects in the ability of CD8⁺ T cells to migrate to the liver, or defects in the viability of CD8⁺ T cells following cDC1 depletion. We have therefore amended the discussion to mention this possibility (*p 19, lines 531-537*).

Comment 10: *Discussion: Several references are missing from the discussion, e.g. "... challenge the paradigm that cognate DC licensing is restricted to SLOs"; "The portal interstitium surrounds the branches of the portal vein, hepatic artery and biliary tree that extend throughout the entire liver; thus, it occupies a significant volume and represents a vast compartment that extends throughout this large solid organ". All the last 3 paragraphs are almost not referenced at all.*

Response: Our apologies for this omission. We have added the appropriate references to the discussion.

Reviewer 3:

Comment 1: *Fig.1: Anti-CD4 treatment depletes CD4⁺ T cells and possibly MAIT cells; reconstitution experiments using gBT-1 CD8 cells together with CD4 T cells or MAIT cells would clarify this issue. The authors also need to show that "intrahepatic CD8⁺ cells" are not MAIT cells (MAIT cells include a CD4 CD8 double positive cell population).*

Dynamics of CD8 activation (CD25, 69, IFN γ , CD107a) and proliferation (Ki67) is not clear in the Figure 1 and 2. The authors should indicate the percentage and number at each time point (D0-D46). This will clarify why the authors chose Day 9, 12, and 15 for representative time point in the figure 2 and after.

Response: While MAIT cells represent up to 45% of lymphocytes in the human liver (Dusseaux et al. Blood 2011, 117(4):1250-9), they make up less than 1% of lymphocytes in the normal C57BL/6 mouse liver (Rehimpour et al. J Ex Med 2015, 212(7):1095-108). Within this 1%, only 20% of MAIT cells expressed a CD4 or CD8 coreceptor, whereas ~70% were double negative for CD4 and CD8. Thus, while we cannot totally rule out that a very low number of MAIT cells in the mouse liver plays a significant role in influencing conventional CD8⁺ and/or CD4⁺ T cell responses in the liver, we believe that this is highly unlikely in our model considering their low frequency in the mouse liver and that double positive T cells did not expand in the liver following rAAV inoculation (Figure shown below). As our model relies on the generation conventional TCR Tg CD8⁺ and CD4⁺ T cells that recognise peptide antigens presented on MHCI and MHCII molecules respectively, the most likely model is that CD4 help is mediated by the large number conventional peptide-MHCII specific CD4⁺ T cells that expand in our model.

The term "intrahepatic CD8⁺ T cells" refers to conventional CD8⁺ cytotoxic T cells recognising peptide antigens in the context of MHCI molecules. In order to increase clarity for the reader,

we have highlighted the fact that we are investigating the effects of CD4 help on conventional CD8⁺ cytotoxic T cell responses in the liver throughout the introduction and results.

Figure: Proportion of CD4⁺CD8α⁺ liver lymphocytes post rAAV inoculation.

Recipient mice received an adoptive transfer of 1×10^4 CD45.1⁺ naïve gBT-1 cells one day prior to treatment with rAAV^{gB-Ag85b}, and the proportion of CD4⁺CD8α⁺ lymphocytes were determined in the livers of rAAV treatment mice 5-, 7-, 9-, 11- and 15-days post rAAV. (A) Representative FACS plot and (B) quantification of the proportion (%) of CD4⁺CD8α⁺ lymphocytes of total live liver lymphocytes. Data is representative of n = 3-4 mice from two independent experiments.

CD25 and CD69 upregulation was used as a readout of very early activation (day 0 - day 3) as performed in similar studies assessing the response of CD8⁺ and CD4⁺ TCR Tg T cells to viral infection in mice (*Hor et al. Immunity 2015, 43, 3:554-65; Eickhoff et al. Cell 2015, 162, 6:1322-37*).

By day 6, all CD8⁺ gBT-1 T cells are activated and have undergone several rounds of cell division in our model, as indicated by CFSE dilution (**Figure 2C**). To gain an idea of the dynamics of the CD8⁺ T cell response with and without CD4 help, we have performed a kinetic experiment in which we measured T cell numbers over 2-day increments from day 5 to day 15 post rAAV inoculation (**Figure 2D and 2E**). The analysis of this kinetic revealed that day 9 was a critical time point when intrahepatic CD8⁺ T cell numbers start to increase in a help dependent manner. Three days later, (at day 12), CD8⁺ T cells numbers were significantly increased while at day 15, CD8⁺ T cells acquired effector function (**Figure 2D, 2E; Figure S2A, S2B**). We have added quantification of the percentage of CD107a⁺IFN-γ and CD107a⁺IFN-γ⁺ gBT-1 T cells at day 15 post rAAV (**Figure S2C**).

Comment 2: *The interpretation of these panels is perplexing. The representative panels shown in Figures 3E and F would seem to indicate a predominance of CD45.1+ tdT- CD8 cells in both PV and CV regions. However, if CD45.1+ tdT- and CD45.1- tdT+ CD8 cells have identical or similar characteristics, the ratio of CD45.1+ tdT- to CD45.1- tdT+ CD8 cells should remain 1:9 across the liver. The authors show low magnification images to help the reader understand the frequency of these cell clusters in the PV and CV regions. The ratios of CD45.1+ tdT- and CD45.1- tdT+ CD8 cells should also be summarised for all ROIs in the liver. Flow cytometric analysis of CD8 T cells throughout the liver would also be helpful.*

Response: We have performed a nearest neighbour analysis of CD45.1⁺ and tdTomato⁺ gBT-1 T cells in portal tracts, PCV regions and the sinusoidal compartment of the liver to both

provide quantification and to show that the ratio of cells in the sinusoids, as a whole, resembles the original adoptive transfer ratio of 9,000 tdTomato⁺ to 1,000 CD45.1⁺ gBT-1 T cells (**Figure 3H**). We have also added flow cytometric analysis of the ratio of tdTomato⁺/CD45.1⁺ gBT-1 T cells in the inguinal lymph nodes (ILN) of mice that received an initial adoptive transfer of 1,000 CD45.1⁺ and 9000 tdTomato⁺ gBT-1 T cells, 15-days post rAAV treatment to show that the initial ratio is preserved as a whole following expansion and recirculation of gBT-1 T cells (**Figure S3E**). To increase clarity for the readers, we have amended the results section and added a new paragraph describing these results (*p 8,9, lines 207-224*).

Comment 3: *CD40 is expressed on a variety of APCs, including Kuepfer cells, B cells and hepatic stellate cells. The authors should deplete these APCs and clarify why the CD40-40L interaction between T and cDC1 is important and not other APCs.*

Response: We believe that our results showing CD4⁺ T cell mediated upregulation of CD40 expression on liver cDC1s and the results of the cDC1 depletion experiments (**Figure 7D-7I**), together with several recent publications showing that CD40 expression on cDC1s is essential for the transfer of help via cognate DC licensing when antigens are cross-presented (*Hor et al. Immunity 2015, 43, 3:554-65; Eickhoff et al. Cell 2015, 162, 6:1322-37; Ferris et al. Nature 2020, 584, 7822:624-29*), provide sufficient evidence of the involvement of CD40 expression by cDC1 in the CD4⁺ T cell mediated intrahepatic expansion of CD8⁺ T cells responding to antigens expressed by hepatocytes. While our results suggest that CD40-CD40L interactions and cDC1s play a dominant role in promoting the help dependent expansion of intrahepatic CD8⁺ T cells (CD40 expressing non-XCR1⁺ APCs cannot compensate completely for depletion of XCR1⁺ cells), we agree that we cannot rule out any contribution of CD40-CD40L interactions involving other liver APCs in promoting an expansion of intrahepatic CD8⁺ T cells. However, while we agree that this is an interesting and worthy area of investigation, we feel that this question, which is an area of ongoing query, is out of the scope of the current work. We have toned down some of our conclusions and have added a new paragraph in our discussion to mention these alternative explanations. (*p17, lines 457-461*).

Comment 4: *What types of CD4+ T cells are critical in activation and proliferation of hepatic CD8+ cells in this system? Fig. 4 shows that almost a week is required to detect CD4+ T cells originated from adoptively transferred naïve CD4+ T cells. However, it is possible that both rapidly activated and differentiated cell and activated tissue resident cd4+ T cells can enhance this process. The authors should investigate the kinetics of CD4 and CD8 in the mice reconstituted with mixed bone marrow of gBT-1 and P25 mice and treated with the AAV.*

Response: Due to the nature of the model, in which T cells are responding to peptide antigens initially expressed by hepatocytes that are presented in the context of MHC I and MHC II molecules, it is likely that CD4 help will be provided by conventional peptide-epitope specific CD4⁺ T cells. Our results confirm that conventional CD4⁺ T cells are robustly activated in the liver draining lymph nodes and form a memory population following the response to a model antigen expressed by hepatocytes (**Figure S1D and S1F**), as demonstrated in our previous work (*Tay S et al. J Immunology 2014 193: 2087*). The results of experiments in which we deplete CD4 T cells at different time points (presented in Figures 4 and S4), suggest that help signals provided by rapidly activated and differentiated CD4⁺ T cells before 9-days post rAAV inoculation do not significantly contribute to the late (post day 9) intrahepatic expansion of CD8⁺ T cells in our model (**Figure 4A-4C; Figure S4A and S4B**). We also have performed all our experiments in naïve (gB-Ag85b antigen inexperienced) mice, and thus there should be no pre-activated CD4⁺ T cells specific against the gB-Ag85b expressed by hepatocytes, or antigens contained within the rAAV capsid, including tissue resident memory CD4⁺ T cells. While we cannot rule out a contribution from bystander CD4⁺ T cells, our results showing that CD40-CD40L interactions are required (**Figure 5A and 5B**), as well as that cDC1s are presenting both gB peptide and peptide 25 (recognised by gBT-1 and P25 T cells,

respectively) (**Figure 7B and 7C**), suggest that the dominant mechanism is cognate antigen-specific.

While we agree that the role of memory CD4⁺ T cell help in primary and memory CD8⁺ T cell responses against antigens expressed in the liver is an important area of study, we believe this is outside of the scope of the current study.

Minor comment 1: *Un-transferred control data enables readers to better understand antigen specificity in proliferation and activation of CD8⁺ cells in the liver.*

Response: The aim of this study was to investigate the hepatocyte-expressed “antigen-specific” CD8⁺ T cell response in the liver. In each of our experiments involving CD8⁺ T cell activation and proliferation, we have used gBT-1 CD8⁺ TCR Tg T cells. This lineage is on a RAG1^{-/-} background and thus all CD8⁺ T cells express a single TCR, excluding the presence on studied cells of dual TCRs that may recognise alternative antigens. As we are able to specifically track this population, we have been able to ensure the antigen specificity of the population studied. Of note, our results in **Figure S1** show that adoptively transferred gBT-1 T cells, as well as P25 T cells, do not become activated in mice not treated with rAAV^{gB-Ag85b} (**Figure S1C and S1D**), indicating that antigen recognition is required.

Minor comment 2: *AAV is infected in relatively small portion of cells. To eliminate the possibility of non-specific activation of hepatic CD8⁺ T cells, the author should show the production of IFN γ and CD107a of host CD45.2 CD8⁺ T cells under CD4-full or –depleted conditions. In addition, the proportion of IFN γ ⁺ and CD107a⁺ CD8 cells seems comparable with or without anti-CD4 treatment. The changes in the numbers and proportions of CD8, IFN γ /CD107a CD8 cells in the liver and spleen from both host and donor cells are necessary. It is helpful for readers to understand the whole picture of this system.*

Response: Endogenous CD8⁺ cytotoxic T cells specific for the gB peptide presented on H2-Kb also exist in C57BL/6 mice (*Mueller et al. Immunol Cell Biol. 2002, 80, 2:156-63*). These cells could respond to rAAV treatment along with the adoptively transferred TCR Tg cells, as well as bystander T cells. Thus, to ensure we are measuring an epitope specific CD8⁺ cytotoxic T cell response, and not a possible bystander responses, we have focused our analysis on CD8⁺ gBT-1 T cells that have defined specificity.

Minor comment 3: *The authors show the image of lower magnifications for providing the frequencies of T cell accumulation in the whole liver in Fig.3, 4, 5, and 6.*

Response: Appropriate lower magnification images have been added (**Figure S5-S7**).

Minor comment 4: *GFP⁺ hepatocytes are attacked and eliminated by activated gBT-1 CD8 T cells? The kinetics of GFP⁺ cells is also an interesting outcome of activated and proliferated CD8 T cells.*

Response: We have analysed of the number of GFP⁺ hepatocytes in the livers of mice depleted of CD4⁺ T cells, or treated with isotype control, at Day 16 post rAAV treatment. The results of this analysis show that the clearance of GFP⁺ antigen expressing hepatocytes is increased when CD4 help is present (**Figure S2C and S2D**).

Minor comment 5: *In typical cases, the liver diseases are categorised by the pathological dominancy, particularly periportal or pericentral dominancy, but this model affects both zone 1 and 3. The levels of AST and ALT may help understanding the main focus of inflammation in the liver. In addition, the authors should discuss what types of liver damage or what liver diseases this model represents.*

Response: As the model employed in this study involves the de novo expression of antigen by hepatocytes, it likely reflects early events in infections with hepatotropic viruses such as HBV and HCV. However, it is important to note the vector used in these studies is non-replicative, and furthermore, transduction occurred in only a small minority of hepatocytes. The source of foreign antigen was thus finite and limited, and the degree of inflammation induced in the associated CD8⁺ T cell response also limited. The aim of the current study was to model early events in the intrahepatic CD8⁺ T cell response, and the role of CD4 help in shaping this response, rather than to develop a disease or pathogenesis-specific model. We would certainly agree that further work developing models based on this data that more fully reflect hepatotropic infection are warranted, however such work will require the development of replicative vectors. To clarify this issue, we have amended the discussion (*p18, lines 507-512*).

REVIEWER COMMENTS

Reviewer #1 (Remarks to the Author):

satisfaction

Reviewer #2 (Remarks to the Author):

The authors have experimentally addressed almost all the points and convincingly argued their case for everything else. Congratulations on this very interesting manuscript.

Just one final suggestion. It would be ideal to have a better clarification of how the quantification of the IF (for example, in Fig. S4F and S5B and C. Please do also for all the other Ifs.) was done. What does each dot represent? How many ROIs, regions, or fields per mouse have been quantified? Reporting the figure number on top of each figure would have been helpful.

Reviewer #3 (Remarks to the Author):

The authors provide some additional data that successfully support their conclusions. This reviewer raises a few issues related to the previous comments.

sFig.2C shows significant reduction in CD107a+IFN γ - CD8 T cells and suggests that the depletion of CD4 T cells may alter characteristics of intrahepatic CD8 T cells. Therefore, these data do not fully support authors' claims, "CD4 help increased the pool of functional memory CD8+ T cells ... rather than increasing the proportion of cells acquiring intrinsic effector function." The depletion of CD4 T cells clearly reduces the quantity of CD8 T cells, but the authors should carefully describe whether CD4 T cells affect the quality of intrahepatic CD8 T cells by analysing the proportion from D0 to D46 over time.

The new panels in Fig.3 and sFig.3 supports the authors' claim that CD4 T cells help local expansion of CD8 T cells in PV and PCV regions. Although local expansion would change the original 1:9 CD45.1/tdTomato ratio in both increasing and decreasing directions, CD45.1+ T cells preferentially

proliferate in the liver. The author should carefully eliminate the possibility that Fig.3H results from expansion defects in tdTomato+ gBT-1 cells.

As the authors (Fig.7H) and other groups (Hor et al. *Immunity* 2015, 43, 3:554-65; Eickhoff et al. *Cell* 2015) presented, cDC1 is important in activation of CD8 T cells. In addition, this reviewer recognises that CD4 T cells, CD40L, CD40, and cDC1 contribute to this process. However, without data from cell-specific deletion of CD40 or CD40L (Ferris et al. *Nature* 2020, 584, 7822:624-29), it remains unclear whether the CD40-40L interaction between CD4 T and cDC1 is important in the liver. Indeed, the reduction in CD40 expression in cDC1 is mild (Fig. 7G), despite a profound reduction in CD8 T-cell numbers in CD4 T-depleted mice. This differs from the situation in *Xcr1-cre/CD40-flox* mice (Ferris et al. *Nature* 2020, 584, 7822:624-29). Since one of many strengths of this paper is that it demonstrated a tissue-specific immune response in the liver during inflammation, the authors should show significant reduction of CD40 on cDC1 in the liver of CD4 T-depleted mice by microscopy or spatial transcriptome analysis and impaired proliferation of CD8 T cells in the presence of anti-CD40 neutralising antibody in Fig.7D. Otherwise, it is not fair to state that the CD40-CD40L interaction between CD4 T and cDC1 plays a dominant role in intrahepatic CD8+ T cell proliferation.

Minor comments:

In page 9 line 220, gbT-1 should be gBT-1.

Point by point response to the reviewers:

Reviewer #1:

satisfaction

Reviewer #2:

The authors have experimentally addressed almost all the points and convincingly argued their case for everything else. Congratulations on this very interesting manuscript.

Just one final suggestion. It would be ideal to have a better clarification of how the quantification of the IF (for example, in Fig. S4F and S5B and C. Please do also for all the other Ifs.) was done. What does each dot represent? How many ROIs, regions, or fields per mouse have been quantified? Reporting the figure number on top of each figure would have been helpful.

Response: Specific details related to image quantification and explanations of what each data point represents have been added to all appropriate figure legends to make clear to readers (Figure legends S3F, S4F, S5B-S5C, 7D and 7E).

Reviewer #3:

The authors provide some additional data that successfully support their conclusions. This reviewer raises a few issues related to the previous comments.

sFig.2C shows significant reduction in CD107a+IFN γ - CD8 T cells and suggests that the depletion of CD4 T cells may alter characteristics of intrahepatic CD8 T cells. Therefore, these data do not fully support authors' claims, "CD4 help increased the pool of functional memory CD8+ T cells ... rather than increasing the proportion of cells acquiring intrinsic effector function." The depletion of CD4 T cells clearly reduces the quantity of CD8 T cells, but the authors should carefully describe whether CD4 T cells affect the quality of intrahepatic CD8 T cells by analysing the proportion from D0 to D46 over time.

Response: We agree that while our findings support the conclusion that CD4⁺ T cells increase the total number of effector and memory transgenic CD8 T cells in liver and spleen at all time points examined (day 15 and 46, see Fig. 1F, S), they have some influence on the proportion of functional CD8⁺ T cells at day 15 (Fig. 2C) but not at day 46 (Fig1F). Based on this difference at day 15, we agree with the reviewer that our claim that CD4⁺ T cells do not influence the effector function of effector and memory CD8⁺ T cells is incorrect. While analysing the proportion of transgenic CD8 T cells activated in the absence or presence of CD4⁺ T cells from D0 to D46 over time might reveal a transient influence of CD4⁺ T cells on the CD8⁺ T cell function, it would not change our conclusion that the overall effect of CD4 T cells is to expand the total number of functional CD8⁺ T cells.

Nevertheless, the reviewer's comment regarding Fig. 2C is pertinent and valid. To address this, we have changed our conclusion to highlight the key role of CD4⁺ T cells on the total number of effector and memory CD8⁺ T cells without mentioning their potential influence on intrinsic CD8⁺ T cell effector function. Our conclusion now reads: "Together, these findings indicated that CD4 help plays an important role in expanding the total number of functional memory CD8⁺ T cells following the response to hepatocyte-expressed antigen."

The new panels in Fig.3 and sFig.3 supports the authors' claim that CD4 T cells help local expansion of CD8 T cells in PV and PCV regions. Although local expansion would change the original 1:9 CD45.1/tdTomato ratio in both increasing and decreasing directions, CD45.1+ T

cells preferentially proliferate in the liver. The author should carefully eliminate the possibility that Fig.3H results from expansion defects in tdTomato+ gBT-1 cells.

Response: Before addressing this comment, we would like to clarify that the experiments shown in Fig. 3 and S3 aim to demonstrate that TCR transgenic CD8⁺ T cells undergo local expansion within portal tracts and surrounding central veins by quantifying their distribution rather than the total number of CD8⁺ T cells. If CD45.1⁺ or tdTomato⁺ T cells proliferated at the same rate, we predict that they would maintain their initial 9:1 ratio and be randomly distributed in the blood and liver sinusoids. If the large number of T cells detected in portal tracts and surrounding central veins resulted from local expansion of T cells reaching these areas, we would expect that it would lead to the formation of clonal cell clusters of CD45.1⁺ and tdTomato⁺ T cells. In contrast, if the large number of periportal T cells resulted from the solely recruitment of circulating T cells, we would expect these T cells to be randomly distributed in these areas.

The images shown in Fig 3F, 3G, S3C and S3D were selected to illustrate some examples of CD45.1⁺ transgenic CD8⁺ T cell clusters in portal tracts and central veins despite being 9 times less frequent in the initial inoculum than tdTomato⁺ transgenic CD8⁺ T cells. Their clonal distribution was supported by quantifying the fraction of nearest non-self neighbours in a large number of PTs from several images (Fig. 3H).

While these images support the local expansion model, they could mislead the reviewer to think that CD45.1⁺ T cells are enriched in portal tracts while tdTomato⁺ T cells are underrepresented. However, this is not the case when looking at additional portal tract and central vein images where CD45.1⁺ T cells were rarer than tdTomato⁺ T cells. To avoid any misinterpretation of these images, we have added some clarifying sentences to the results section and have provided additional representative images of portal tracts containing gBT-1 T cell clusters showing that CD45.1⁺ CD8 T cell clusters do not predominate (Supplementary figure 3e). To address the reviewer comments, we have added quantification of the total number of tdTomato⁺ and CD45.1⁺ gBT-1 T cells in portal tracts, pericentral venous regions and sinusoidal regions (Fig. S3F). This new quantification shows that the ratio of CD45.1⁺:tdTomato⁺ gBT-1 T cells in all compartments of the liver resemble the initial adoptive transfer ratio, arguing against preferential proliferation advantage of CD45.1⁺ T cells over tdTomato⁺ T cells. These findings thus support a model in which CD45.1⁺ gBT-1 T cells clonally expand in portal tracts and PCV regions. The text of the results section included in the revised manuscript has been changed accordingly.

As the authors (Fig.7H) and other groups (Hor et al. Immunity 2015, 43, 3:554-65; Eickhoff et al. Cell 2015) presented, cDC1 is important in activation of CD8 T cells. In addition, this reviewer recognises that CD4 T cells, CD40L, CD40, and cDC1 contribute to this process. However, without data from cell-specific deletion of CD40 or CD40L (Ferris et al. Nature 2020, 584, 7822:624-29), it remains unclear whether the CD40-40L interaction between CD4 T and cDC1 is important in the liver. Indeed, the reduction in CD40 expression in cDC1 is mild (Fig. 7G), despite a profound reduction in CD8 T-cell numbers in CD4 T-depleted mice. This differs from the situation in Xcr1-cre/CD40-flox mice (Ferris et al. Nature 2020, 584, 7822:624-29). Since one of many strengths of this paper is that it demonstrated a tissue-specific immune response in the liver during inflammation, the authors should show significant reduction of CD40 on cDC1 in the liver of CD4 T-depleted mice by microscopy or spatial transcriptome analysis and impaired proliferation of CD8 T cells in the presence of anti-CD40 neutralising antibody in Fig.7D. Otherwise, it is not fair to state that the CD40-CD40L interaction between CD4 T and cDC1 plays a dominant role in intrahepatic CD8+ T cell proliferation.

Response: We have performed two experiments requested by the reviewer.

(i) cDC1s were sorted from the livers of rAAV treated mice at 12-days post rAAV treatment, following treatment with isotype control or anti-CD40L blocking mAb. Sorted liver cDC1s were

co-cultured with naïve CTV-labelled gBT-1 T cells at a cDC:T cell ratio of 1:10 for 72 hours and the level of activation and proliferation of gBT-1 T cells was assessed. Proliferation of CD8 T cells from animals treated with CD40L blocking antibody was impaired, further demonstrating that CD40-CD40L interactions on liver cDC1s were required for the optimal expansion of CD8⁺ T cells. The result of this additional experiment is now summarised in 2 new panels in **Fig. 7 (Figure 7H, 7I)**. A paragraph explaining the experiments and summarising the findings has also been added to the results section.

(ii) CD40 expression by liver cDC1s was assessed in situ in portal tracts, via imaging of livers from mice administered rAAV 12-days prior and treated with isotype control or anti-CD4 depleting mAb. Liver sections from recipient mice were stained with anti-CD40 mAb. The images (summarised in a new panel of Fig. S7 (**Figure S7D**), show that amongst MHCII⁺ cells detected in portal tracts of mice containing CD4 T cells, some hepatic cDC1s expressed higher levels of CD40 expression than other MHCII⁺ cells. In contrast, in mice lacking CD4 T cells, all hepatic cDC1s expressed similar low levels of CD40. Quantification of the MFI of CD40 expression on cDC1s in the presence and in the absence of CD4 T cells (**Figure S7E**) confirmed these findings. These experiments suggest that CD40 is upregulated by liver cDC1s in situ within portal tracts in the presence of CD4 help.

Together, we believe these experiments strengthen our conclusions that CD4⁺ T cells license liver cDC1s via CD40L-CD40 interactions in situ in portal tracts, which in turn enhances the expansion of intrahepatic CD8⁺ T cells responding to hepatocyte expressed antigen.

Minor comments:

In page 9 line 220, gbT-1 should be gBT-1.

Response: We have corrected this formatting error.

REVIEWERS' COMMENTS

Reviewer #2 (Remarks to the Author):

Thanks for clarifying this aspect.

Reviewer #3 (Remarks to the Author):

The authors have conducted additional experiments in accord with my comments, the results of which satisfactory answer to this reviewer's previous queries.